# Consequences of Disturbing Manganese Homeostasis

**DOI:** 10.3390/ijms241914959

**Published:** 2023-10-06

**Authors:** Jacek Baj, Wojciech Flieger, Aleksandra Barbachowska, Beata Kowalska, Michał Flieger, Alicja Forma, Grzegorz Teresiński, Piero Portincasa, Grzegorz Buszewicz, Elżbieta Radzikowska-Büchner, Jolanta Flieger

**Affiliations:** 1Chair and Department of Anatomy, Medical University of Lublin, 20-090 Lublin, Poland; wwoj24@wp.pl (W.F.); aforma@onet.pl (A.F.); 2Department of Plastic, Reconstructive and Burn Surgery, Medical University of Lublin, 21-010 Łęczna, Poland; aleksandrabarbachowska@gmail.com; 3Department of Water Supply and Wastewater Disposal, Lublin University of Technology, 20-618 Lublin, Poland; b.kowalska@pollub.pl; 4Chair and Department of Forensic Medicine, Medical University of Lublin, 20-090 Lublin, Poland; michalflieeeger@gmail.com (M.F.); grzegorz.teresinski@umlub.pl (G.T.); grzegorz.buszewicz@umlub.pl (G.B.); 5Clinica Medica A. Murri, Department of Biomedical Sciences & Human Oncology, Medical School, University of Bari, 70124 Bari, Italy; piero.portincasa@uniba.it; 6Department of Plastic, Reconstructive and Maxillary Surgery, CSK MSWiA, 02-507 Warszawa, Poland; elzbieta.radzikowska@gmail.com; 7Department of Analytical Chemistry, Medical University of Lublin, 20-093 Lublin, Poland

**Keywords:** manganese, exposure to manganese, manganese toxicity, manganese neurotoxicity

## Abstract

Manganese (Mn) is an essential trace element with unique functions in the body; it acts as a cofactor for many enzymes involved in energy metabolism, the endogenous antioxidant enzyme systems, neurotransmitter production, and the regulation of reproductive hormones. However, overexposure to Mn is toxic, particularly to the central nervous system (CNS) due to it causing the progressive destruction of nerve cells. Exposure to manganese is widespread and occurs by inhalation, ingestion, or dermal contact. Associations have been observed between Mn accumulation and neurodegenerative diseases such as manganism, Alzheimer’s disease, Parkinson’s disease, Huntington’s disease, and amyotrophic lateral sclerosis. People with genetic diseases associated with a mutation in the gene associated with impaired Mn excretion, kidney disease, iron deficiency, or a vegetarian diet are at particular risk of excessive exposure to Mn. This review has collected data on the current knowledge of the source of Mn exposure, the experimental data supporting the dispersive accumulation of Mn in the brain, the controversies surrounding the reference values of biomarkers related to Mn status in different matrices, and the competitiveness of Mn with other metals, such as iron (Fe), magnesium (Mg), zinc (Zn), copper (Cu), lead (Pb), calcium (Ca). The disturbed homeostasis of Mn in the body has been connected with susceptibility to neurodegenerative diseases, fertility, and infectious diseases. The current evidence on the involvement of Mn in metabolic diseases, such as type 2 diabetes mellitus/insulin resistance, osteoporosis, obesity, atherosclerosis, and non-alcoholic fatty liver disease, was collected and discussed.

## 1. Introduction

Chemical elements are divided into “macroelements”, “trace elements”, and “ultra-trace elements” according to their content in the human organism. Chemical elements, such as oxygen (O), carbon (C), hydrogen (H), nitrogen (N), calcium (Ca), phosphorus (P), potassium (K), sodium (Na), sulfur (S), chlorine (Cl), and magnesium (Mg), occur in concentrations greater than 0.01%; a group of trace elements, such as Fe, Zn, fluorine (F), strontium (Sr), molybdenum (Mo), Cu, iodine (I), Mn, and boron (B) range from 0.00001% to 0.01%; while ultra trace elements are present at concentrations lower than 0.00001% [1].

Despite their low content, trace elements are important in physiological processes that regulate all vital functions of the organism. Each trace element has an optimal concentration range. Therefore, any disturbance in their homeostasis, namely hypo (micro)elementosis or hyper(micro)elementosis, causes a state of stress and contributes to the development of diseases. However, it should be emphasized that the role of trace elements in the development of many diseases is not yet fully understood, and new data on this subject are regularly emerging [2,3,4,5,6,7,8,9,10,11,12,13,14,15].

Trace element deficiencies are often the result of inadequate dietary intake. The excessive accumulation of trace elements in the human body is usually associated with unfavorable environmental conditions, occupational settings, or even genetic factors. Various factors, such as geographical location, the type of job, and dietary habits, affect the levels of trace elements in human tissues, so the ranges measured may differ between the populations studied [2].

In terms of their physiological role, chemical elements, such as Fe, I, Cu, Zn, Co, chromium (Cr), Mo, selenium (Se) and Mn are, “structural” essential trace elements. This means that a reduction in their concentration in the body below a certain limit results in a disturbance of a physiologically important function [16]. Recommended dietary reference intakes for essential trace elements have been established by the US National Academy of Sciences, Food and Nutrition Board [17].

Chemical elements are found in water, for example, the macroelements Ca, Mg, K, Na, Fe, Mn, and Cu, and in soils in the form of silicates (Mn and Zn), sulfides (Cu, Fe, and Zn), oxides (Fe_2_O_3_ and Fe_3_O_4_), and the native elemental form (Cu and noble metals). Se, a non-metal, one of the trace elements, can occur in the form of an element; toxic inorganic compounds, such as selenate and selenite; as well as organic compounds, for example with polysaccharides and proteins, which are biologically active and good sources of Se. In turn, Fe, the second most abundant metal in the Earth’s crust and one of the most important elements for all living organisms due to, among other things, the possible interconversion of Fe^2+^ to Fe^3+^ in redox systems, is rarely found in its native elemental form, except meteorites. Instead, it is common in a dissolved form in water, as well as in minerals, such as pyrite (FeS_2_) and siderite (FeCO_3_). Food components are linked to a geological issue through the food–soil–rock chain [18]. The cereal diets available to people in disadvantaged areas, or meals prepared from highly processed foods, do not provide sufficient minerals. To compensate for the lack of essential minerals, manufacturers produce so-called fortified foods.

In the body, alkali metals, such as Na^+^ and K^+^, occur as hydrated ions with high mobility. These ions have weak ligand binding strengths that are predominantly ionic in origin. Mg^2+^ and Ca^2+^ have intermediate binding strengths to organic ligands, which are most commonly coordinated by oxygen donors derived from glutamic acid, aspartic acids, carbonyl oxygens, peptides, and water. In turn, transition metal elements containing electrons in the d-shell form metal complexes with strongly bound ligands. For example, Fe, which is mostly bound in hemoglobin, ferritin, myoglobin, and transferrin, is also a component of many enzymes, e.g., catalase, oxidases, peroxidases, dehydrogenases, cytochrome, or nucleotide reductase; Cu is bound to several proteins (cerebrocuprein, erythrocuprein, and hepatocuprein) and enzymes (tyrosinase, cytochrome oxidase, ascorbic acid oxidase, uricase, ceruloplasmin, superoxide dismutase, amine oxidase, and dopamine hydroxylase); and Zn is an essential component of many metalloenzymes (carbonic anhydrase, alkaline phosphatase, pancreatic carboxy-peptidases, erytosoly superoxide dismutase, and relinene reductase).

Mn is an essential trace mineral involved in energy metabolism and the regulation of brain and nerve function. Figure 1 shows a schematic of the primary sources of Mn exposure and its role in the human body.

However, when accumulated in large amounts, Mn is toxic to cells as it increases oxidative stress, impairing mitochondrial function and leading to cell apoptosis. Precise homeostatic mechanisms are required to maintain adequate levels of intracellular Mn. Cells have transport mechanisms for the uptake, intracellular distribution, and efflux of Mn ions. Figure 2 shows selected consequences of Mn dyshomeostasis.

Both Mn deficiency and Mn excess can lead to adverse health outcomes. There tends to be a U-shaped relationship (Figure 3) between the risk of adverse effects and Mn intake.

The consequences of disrupting Mn homeostasis are dangerous for the central nervous system, especially for the function and development of the brain, which selectively accumulates this element. Excessive exposure to Mn has been repeatedly associated with an increased risk of behavioral disorders in children and neurodegenerative diseases in adults. It should be noted that Mn accumulation may be the result of not only environmental exposure, but also genetic disorders of Mn metabolism.

The neurotoxicity of Mn is a consequence of changes at the molecular level. In order to monitor risks, it is necessary to establish reference ranges for biomarkers of Mn in different matrices. These issues are still in progress and require an interdisciplinary approach involving medicine, biology, and analytical chemistry. The results of experimental research should be systematically collected to facilitate the design of future studies and strategies for protection against Mn-induced toxicity, the biomonitoring of Mn exposure, and the updating of daily intake.

We conducted a comprehensive search of the Pubmed database from 2018 to 2023. A preliminary selection was performed based on the title and abstract of the article, with the aim of providing an overview of the state of knowledge on the role and homeostasis of Mn, as well as current research highlighting the problems to be solved, and challenges and trends that may shape the future research. A total of 13,961 scientific papers on “Manganese” appeared in the Pubmed database, of which 761 were review articles, less than half of which, 443, were reviews on the importance of this element for the brain. Only 27 clinical trials on Mn were published between 2018 and 2023. These included studies of Mn superoxide dismutase (MnSOD) and oxidative stress in relation to atopic dermatitis, enteral nutrition, and psychiatric disorders with a schizophrenic episode. Some of the research concerned the involvement of Mn in glucose metabolism, carcinogenesis, and wound healing. Another group of studies involved the quantitative determination of Mn in biological samples, such as hair, plasma, and umbilical cord blood, in order to correlate Mn levels with cognitive performance, the type of diet, and the monitoring of drug therapy.

The aim of this review is to present the current state of knowledge on the role of Mn in the body, including the sources and routes of Mn exposure, reference levels in different samples, the mechanism of action, organs that preferentially accumulate Mn, the competitiveness of Mn with other metals, and its role in disease development, particularly in brain pathology, infectious diseases, metabolic diseases, fertility, and wound healing. The review cited both papers collected from the Pubmed database and relevant articles that were listed in their references. A total of 563 references were selected and discussed.

## 2. Manganese (Mn)

Mn is a “structural” essential trace mineral involved in many different biological processes, including energy metabolism, antioxidant function, detoxification, musculoskeletal, immune and reproductive systems, bone development, and the regulation of brain and nerve function. Mn is a cofactor for numerous cellular enzymes [19] involved in carbohydrate, nitrogen, oxygen radical, glycosaminoglycan, and cholesterol metabolism.

Mn has a high redox activity. There are eleven known oxidation states of Mn, ranging from −3 to +7 with different coordination geometries, of which Mn^2+^ and Mn^3+^ are the most physiologically relevant states. Mn^2+^ is the stable and more easily degraded form in biological systems. However, it is important to note that Mn^3+^, when complexed with transferrin, is a more potent oxidant that is even more stable than Mn^2+^. It should be noted that Mn in the oxidation state Mn^6+^ and Mn^7+^ are highly oxidizing and therefore toxic.

### 2.1. The Source of Exposure to Mn

Mn occurs naturally in soil, rocks, and water and is also used in the manufacture of glass, dry cell batteries, and various other industrial applications, mainly in the chemical, textile, and leather industries. Mn is readily bioavailable when consumed in the form of water, plant products, or dietary supplements. Dietary sources of Mn include whole grains, oats, legumes, nuts, cloves, and cinnamon. Mn is relatively abundant in seafood. It is worth noting that one of the world’s most popular beverages, tea, is a very rich source of Mn. Dried tea leaves contain an average of 1.21 to 5.16 mg/kg, depending on the region of the world [20]. In 2003, the Expert Group on Vitamins and Minerals reported that tea was the largest dietary contributor to Mn intake [21]. Mn and ammonium pyrophosphate complex is a pigment used in the manufacture of cosmetics and paints. In the anthropogenic environment, organic forms of Mn are ubiquitous as components of fungicides (maneb and mancozeb), fuel oil additives, and the petrol additive methylcyclopentadienyl Mn tricarbonyl (MMT). Of particular concern is occupational exposure to Mn in industries such as mining, welding, and battery manufacturing. People living near industrial areas may be exposed to elevated levels of Mn.

There are several routes of exposure to Mn, such as chronic oral exposure from contaminated water [22], the inhalation of Mn, particulate matter from ferromanganese plants [23], patients receiving parenteral nutrition [24,25], or direct intravenous administration, e.g., of Mn-contaminated preparations, such as ephedrine [26,27]. There are also genetic defects, which are intrinsic factors that can affect or alter the distribution or function of Mn in the body.

The inhalation of Mn, as opposed to ingestion, is most effective because inhalation exposure bypasses the intestinal and hepatic toxicity control processes [28]. Mn enters the circulation through the nasal mucosa and then the brain through the blood–brain barrier by facilitated diffusion and active transport from the olfactory bulb to the cerebral cortex [29]. According to research by scientists at Wake Forest University in the USA, even long-term exposure to Mn-containing water in the shower may pose a risk of central nervous system (CNS) neurotoxicity [30]. The authors of the study based their findings on the assumption that intranasally administered Mn bypasses the blood–brain barrier and enters the CNS directly via the olfactory route, as demonstrated in animal studies. According to the authors, even 10 years of bathing in Mn-contaminated water results in exposure to aerosol Mn at doses many times higher than those that cause Mn accumulation in the brain of rats (3 and 112 times higher). Aschner [31] pointed out that the above study lacked information on the solubility of Mn in water, the size of the particles in the air aerosol, and its clearance. The authors also did not determine whether it were possible to reach the threshold concentration of Mn toxicity, which is above 100 µM in cells, by taking a shower every day. The cause of Mn exposure is the inhalation [32,33,34,35,36] of air contaminated with fungicides [37,38] and gasoline enriched with MMT [39]. Fillipini et al. [40] showed a close, statistically significant relationship between the level of particulate matter ≤ 10 µm in the air and the concentration of Mn in serum.

A major source of exposure to Mn is drinking water. The cause of water contamination is usually industrial pollution, or sedimentary or igneous rocks (e.g., augite and hornblende). In most cases, Mn occurs naturally together with Fe. The concentration of Mn in water depends on the location. It is usually between 0.0001 and 0.1 mg L^−1^ [41]. The Regulation of the Minister of Health in Poland of 7 December 2017 on the quality of water intended for human consumption, similar to that in the USA, sets the permissible amount of Mn in water at the level of 0.05 mg L^−1^ [42,43].

It should be noted that the health risks associated with exposure to Mn depend on the level and duration of exposure. It is therefore important to monitor blood Mn levels in order to identify exposure to high levels of Mn and to take steps to reduce the risk of toxicity. Biological biomonitoring is used to assess human exposure to environmental toxicants [44]. Biological samples, such as blood, urine, hair, skin, internal organs, etc., are analyzed for the presence of toxins and their metabolites. However, the interpretation of the results obtained is problematic due to the lack of reference ranges for most tissues [45]. Geometric means and selected percentiles of measured Mn blood concentrations (µg L^−1^), urine concentrations (µg L^−1^), and creatinine-corrected urine concentrations (µg g^−1^ of creatinine) for the US population from the National Health and Nutrition Examination Survey, collected between 2011 and 2018, are available specifically at https://www.cdc.gov/exposurereport/data_tables.html (accessed on 3 May 2022).

In many countries, levels of toxic metals, i.e., mercury (Hg), nickel (Ni), and chromium, are monitored only in the case of occupational exposure. However, they are all monitored in the NHANES exposure reports for the US population. However, the development of modern technologies requires the inclusion of other metals in biological monitoring, e.g., tantalum (Ta), silver (Ag), tungsten (W), beryllium (Be), vanadium (V), neodymium (Nd), and scandium (Sc) [46]. In many countries, including the United Kingdom [44,47,48], Belgium [45,49], the Czech Republic [50], and Germany [51], it has been possible to estimate reference ranges for many metals in biological samples of human origin thanks to extensive epidemiological studies. In the United States, a specific program has been dedicated to biological monitoring, namely “The National Report on Human Exposure to Environmental Chemicals” [52]. Of note is the 2014 study of 132 British adults. Urine samples were collected from 82 men (180 samples) and 50 women (100 samples). The urine samples were analyzed for 61 elements and creatinine using a modern and sensitive analytical technique, inductively coupled plasma mass spectrometry (ICP-MS) [53].

### 2.2. Recommended Dietary Intake of Mn

The dietary intake of Mn depends on age, race, sex, geographical location, and dietary habits and ranges from 1.38 mg/day to 6.8 mg/day [54,55,56,57,58]. The recommended daily intake of Mn has not been established, but reasonable intake levels have been derived from observations in healthy individuals [59]. The recommended oral intake of Mn for adult men is 2.3 mg/day; for women, 1.8 mg/day; and for children, depending on age, from 0.003 mg/day for 0–6 months to 2.2 mg/day for 14–18 years. However, pregnant and lactating women should take a higher dose of 2.0 and 2.6 mg/day, respectively. The parenteral dose is 0.06–0.1 mg/day for adults and 1–50 µg/day for children. According to other sources, the optimal reference value for the dietary intake of Mn is 3–5 mg/day [60,61]. The upper tolerable intake for adults is 11 mg per day. However, Mn supplementation with more than 20 mg Mn has been reported in cases of osteoarthritis and osteoporosis [62,63]. The minimum requirement is 10.8 µg per kg body weight, which is approximately 0.74 mg/day. Another source recommends 2.5–5 mg/day [64]. The World Health Organization recommends that the daily intake of Mn for an adult should be between 0.7 and 10.9 mg [65]. Mn is considered toxic if ingested in excess of 40 mg per day.

The absorption of Mn in the gastrointestinal tract, especially in the small intestine, is rather low—up to 5% [60,66]. Only when inhaled can the majority of Mn enter the body [61]. Vitamins B1 and E facilitate the absorption of Mn through the gastrointestinal tract, whereas the excessive consumption of Ca and P inhibits this process [67]. The half-life of Mn in the body is 4–40 days [68]. The absorption, distribution, and excretion of Mn in bile are actively controlled by homeostatic mechanisms. The liver plays a key role in Mn metabolism. The liver is the site of the accumulation and excretion of manganese in the bile (3.6 mg/day). The normal urinary excretion rate is 0.03 mg/day [1].

Research has confirmed that there are differences in the absorption and metabolism of Mn between men and women. It has been shown that women absorb more Mn than men from a diet with the same Mn content, and that the half-life of Mn is shorter in women than in men [69]. In addition, blood Mn concentrations depend on age, sex, and race, as well as location, and range from 1.6 μg L^−1^ to 62.5 μg L^−1^ [70,71]. It is therefore understandable that optimal levels of Mn intake tend to take gender differences into account. Such a difference between the sexes is also confirmed by other studies [72,73,74]. It should be noted that the difference in blood Mn concentrations between men and women decreases with age [75].

### 2.3. Accumulation of Mn in the Brain

The total amount of Mn in the human body is estimated to be 10–20 mg. Mn has been found in the bones, pancreas, kidneys, and adrenal glands. Studies have shown that Mn is also present in the human brain and that its concentration varies in different regions of the brain. Mn accumulates in areas of the brain that are rich in Fe, i.e., around the basal ganglia, namely in the striatum (caudate nucleus, putamen, and nucleus accumbens), globus pallidus, substantia nigra, and hypothalamic nuclei [76]. Magnetic resonance imaging (MRI) has shown that Mn accumulates in the basal ganglia, particularly in the globus pallidus [77,78,79] and in the frontal cortex. The accumulation is visible as a prominent symmetrical high-signal lesion. Studies in primates have shown that inhaled Mn accumulates in the bulbar olfactory organs in addition to the striatum, frontal cortex, and cerebellum [80]. It should be noted that the concentration of Mn in the human brain can be influenced by several factors, including age, sex, diet, and exposure to environmental toxins. Therefore, the absolute quantification values reported by different research groups differ from each other. Another reason for the observed differences is the reporting of Mn mass per wet weight of the test sample (Mn mass/wet weight) instead of the dry weight of the sample after freeze-drying.

The average concentration of Mn in the whole human brain was estimated to be between 5.32 and 14.03 µg g^−1^ [81]. Study performed using animal models estimated that Mn is distributed in the brain in the following order: substantia nigra > striatum > hippocampus > frontal cortex. This occurs in at least a hundred-times-lower concentration range of 0.03–0.07 µg g^−1^ wet tissue weight [82]. Another study showed that Mn concentrations vary widely in different areas of the brain, with the highest levels found in areas of the basal ganglia and globus pallidus, which are associated with motor control [61]. It should be noted that elevated levels of Mn in the basal ganglia are thought to be a major cause of neurological disorders [83]. The study showed that the mean concentration of Mn in the basal ganglia and globus pallidus regions was 3.5 µg g^−1^ wet weight and 5.5 µg g^−1^ wet weight, respectively. According to a ICP-MS/MS examination performed by Baj et al. [84], the Mn content in different brain areas is not uniform but always exceeds a value of 100 ng g^−1^. The affinity of Mn can be ordered as follows: insula > hippocampus > precentral gyrus ~ head of caudate nucleus. In this cross-sectional study, the existence of negative correlations between the brain levels of Mn and Mn levels in the liver was observed. At the same time, the mean Mn content calculated from measurements from 11 brain areas was about three times lower than that determined in the liver (1144/332.8 ng g^−1^). The mean concentration of Mn in the optic chiasm has been estimated to be lower than 100 ng g^−1^ (88.194 ng g^−1^ wet weight) [85], similarly to the meninges, i.e., the dura mater and the arachnoid is estimated to be 81.41 ng g^−1^ wet weight [86].

It should be noted that the content of Mn in the human brain can be influenced by several factors, including age, sex, health status, chronic diseases, diet, exposure to environmental toxins, etc. The above limitations are the reason for the differences between the absolute values of Mn content in the tissues tested, which are reported by different research groups. Another reason for the observed differences is the reporting of Mn mass per wet weight of the test sample (Mn mass/wet weight) instead of the dry weight of the sample after freeze-drying.

### 2.4. Evaluation of the State of Mn

The most common way to assess Mn status is to measure Mn in whole blood, as the concentration of Mn is slightly higher than in serum. Other researchers suggest testing superoxide dismutase activity. Mn exposure is also reflected in the levels of different types of Mn, namely Mn ferritin (Mn-Fer), Mn transferrin (Mn-Tf), Mn citrate (Mn-Cit), and inorganic Mn (Inorg-Mn) [40,87,88]. One difficulty in determining Mn levels may be the high probability of sample contamination. During sampling, contamination may occur from blood collection bottles or stainless steel needles containing trace amounts of Mn. Mn is also subject to intra-individual variability, making it difficult to establish reference values.

### 2.5. Reference Values for Mn Homeostasis

Reliable reference intervals (RIs) and reference values (RVs) are essential for laboratory analysis and play an important role in the interpretation of test results. The International Federation of Clinical Chemistry (IFCC) published guidelines C28-A3 for a reference population and the statistical analysis of data in 2008 [89]. RIs are derived from a reference distribution, usually a 95% interval, and describe a specific population [90]. The calculation of RIs includes parametric and non-parametric calculation methods, outlier detection, partitioning, and confidence intervals. It is generally recommended that the reference interval should include the central 0.95 fractions of the reference distribution (i.e., 95% of the population). Consequently, the lower reference limits are estimated as the 2.5th percentile and the upper limits as the 97.5th percentile of the distribution of test results for the reference population. Reference limits should always be reported with their 90% confidence intervals (CIs). The CI is a range of values that includes the true percentile (e.g., the 2.5th percentile of the population) with a specified probability, usually 90% or 95%, as the “confidence level” of the interval.

Blood levels are a biomarker of Mn status in the body and an indicator of Mn exposure [91]. A 1977 monograph [92] reported that the concentration of Mn in body fluids and tissues was independent of age. In later years, however, significantly elevated plasma Mn concentrations were observed in breast-fed neonates and infants [93,94]. A study by Rükgauer et al. [93] on the serum or plasma of 137 children and 68 adult donors showed that serum Mn concentrations decrease with age. The reason for the higher serum Mn levels in early childhood is thought to be the higher intestinal absorption of Mn in neonates and infants compared to adults. In turn, the possible interaction of iron (Fe) with Mn may cause sex differences in Mn levels. Lower ferritin concentrations, e.g., in women of reproductive age compared with men, are associated with higher blood Mn concentrations in women [11,12,95]. In addition, an increase in mean whole blood Mn concentrations has been reported during pregnancy [96,97,98]. This increase may be related to increased Mn absorption due to the upregulation of iron absorption, particularly in late pregnancy [99,100].

The short half-life of Mn in the blood (<2 h) due to rapid hepatic clearance [101,102,103] makes the blood Mn concentration level an expression of current exposure. The use of hair as a sample for metal trace analysis is advantageous mainly because of the non-invasive nature of the material collection and the stable composition [104,105], which reflects the long-term exposure of humans to metals in the environment [106]. Blood and urine as test samples are characterized by a significant variability in composition, which is regulated by homeostatic mechanisms [107]. The determination of Mn in tissues, hair, or teeth is again related to long-term exposure. For example, the determination of Mn in hair [23,108,109] or in the dentin of deciduous teeth [37,38] has been used to study cognitive deficits in children. Fingernail analysis can also determine long-term exposure for up to one year [110,111]. Due to the high excretion rate of Mn in bile to feces (95%) and a short half-life, urinary Mn concentrations are not recommended as an optimal sample for estimating exposure for Mn assessment.

In practice, Mn is typically analyzed in plasma, whole blood, urine, and hair. Atomic absorption spectrometry (AAS), neutron activation analysis (NAA), inductively coupled plasma optical emission spectrometry (ICP-OES), and inductively coupled plasma mass spectrometry (ICP-MS) are used to determine metals in biological samples. It should be noted that the low levels require advanced detection techniques and increase the possibility of error. According to the “Toxicological profile for manganese” published by the US Department of Health and Human Services Public Health Service Agency for Toxic Substances and Disease Registry, AAS (furnace technique) has a Mn detection limit of less than 1 µg L^−1^ of urine, feces; flameless AAS, less than 0.2 µg g^−1^ of hair; while ICP-AES, less than 0.2 µg g^−1^ of tissue [112]. Standard AAS is cumbersome for the multi-element quantification of metals and metalloids. In recent years, ICP-MS and ICP-OES techniques, which allow for specificity, sensitivity, and the rapid evaluation of small samples, have been increasingly used in clinical and forensic toxicology to monitor heavy metal exposure and in clinical or forensic investigations. Mn is commonly monitored in the treatment of cirrhosis, parenteral nutrition and environmental exposure to Mn. In order to correctly interpret laboratory data from Mn biomonitoring, it is necessary to define the range of reference values in biological samples.

The reference ranges for Mn in different samples vary. As early as the 1980s, it was noted that the reference serum Mn levels reported by different research groups [113] varied over a wide range of concentrations from 0.54 µg L^−1^ [114] to as high as 13 µg L^−1^ [113,115]. The Mayo Clinic laboratories suggest reference values of 4.7 to 18.3 ng mL^−1^ in whole blood [116], which was determined by triple quadrupole inductively coupled plasma mass spectrometry (ICP-MS/MS). Values of 1 to 2 times the upper limit of normal indicate excess exposure to Mn. The main compartment for circulating Mn is the erythrocytes. Therefore, the concentration of Mn in whole blood can be several times higher than in serum. The highest concentration of Mn was found in the liver, with concentrations in the liver of 1 to 1.5 mg kg^−1^ (wet weight). Mn is excreted in feces. Only small amounts are excreted in the urine. The normal range for urinary Mn levels in healthy individuals is usually less than 1 μg g^−1^ creatinine. Morton et al. [53] reported a mean Mn level of 1.3 mmol/mol creatinine in urine samples from occupationally unexposed individuals. This is much lower than the 3.1 mmol/mol creatinine reported by White and Sabbioni [47]. The reference range for hair Mn levels in healthy individuals is generally between 0.05 and 2.5 μg g^−1^. Goulle et al. [117] proposed reference values with median and reference range from the 5th to 95th percentile for whole blood (7.6 µg L^−1^; 5–12.8 µg L^−1^), plasma (1.12 µg L^−1^; 0.63–2.26 µg L^−1^), urine (0.31 µg L^−1^; 0.11–1.32 µg L^−1^), and hair (0.067 ng mg^−1^; 0.016–0.57 ng mg^−1^) based on analyses of approximately 10,000 samples taken from 100 healthy volunteers. Ten years later, the same authors reported slightly modified reference ranges for whole blood (8.6 µg L^−1^; 5.9–13.3 µg L^−1^) and plasma (0.65 µg L^−1^; 0.35–1.08 µg L^−1^) from 106 adult healthy volunteers, whole blood (7.1–128 µg L^−1^) from 54 deceased individuals, and fingernail and toenail (0.36 µg L^−1^; 0.12–2.08 µg g^−1^) from 50 volunteers [118]. Hair Mn levels can be influenced by various factors, such as hair washing frequency, hair color, and age. Reference values for Mn homeostasis may vary depending on the specific biomarker being measured and the population being studied. However, here are some commonly accepted reference values for Mn homeostasis along with their literature sources (Table 1). It is important to note that reference values for Mn homeostasis may also vary depending on the age, sex, and health status of the individual being studied. In addition to Mn concentration in different samples, various Mn-dependent biochemical parameters, such as MnSOD, are also used to diagnose the state of Mn in the body. Limited studies are available on saliva and prolactin as alternative biomarkers of Mn exposure [119,120].

### 2.6. Association of Mn with Other Metals

Mn can interact with other metals either through competition for absorption, transport, or through shared biochemical pathways.

#### 2.6.1. Magnesium (Mg)

Although both Mg and Mn are important for human health, in theory, there is no direct link between their homeostasis. The absorption and regulation of these minerals are controlled by different mechanisms in the body, and deficiencies or excesses of one mineral do not necessarily affect the other. However, experiments using animal models have shown that dietary Mg deficiency can alter Mn metabolism. Studies confirming this relationship have been carried out in Wistar rats that were fed a Mg-deficient diet. Dietary Mg deficiency reduced Mn concentrations in plasma, brain, spinal cord, lungs, spleen, kidneys, testes, bones, and liver, with the exception of the adrenal glands and blood. A positive correlation was also found between liver Mn concentration and pyruvate carboxylase activity [142]. On the other hand, experiments in healthy adults showed that supplemental Mg at a dose of 200 mg/day reduced Mn bioavailability, probably by reducing Mn absorption or increasing its excretion [143].

#### 2.6.2. Iron (Fe)

Fe and Mn homeostasis are closely linked. The main cellular Fe uptake pathway, the transferrin receptor (TfR), has been shown to mediate the uptake of Mn(III). However, its affinity for Fe(III) is higher [144]. Thus, changes in Fe homeostasis may alter Mn homeostasis. DMT1 (NRAMP2) and ferroportin have been shown to be involved in both Fe and Mn transport [32,145].

The mechanisms of Fe absorption are similar to those of divalent metals, such as Mn, Pb, and Cd [146,147]. Thus, dietary Fe deficiency can lead to excess absorption of Mn, among other divalent metals [69,119,148,149,150,151,152,153]. Mn and Fe compete for absorption in the body because they share absorption and transport proteins, including the divalent metal transporter 1, the lactoferrin receptor, transferrin, and ferroportin in the intestine [154].

In animal studies, excess Mn intake has been shown to impair Fe absorption and lead to Fe deficiency and anemia in animals and humans [155]. Conversely, Fe deficiency can increase Mn absorption [156]. Davis et al. [155] confirmed that Fe supplementation at 60 mg/day for several months caused a reduction in blood Mn levels and decreased MnSOD activity in leukocytes. Many examples of the relationship between Fe and other metals have been collected in the review, confirming that Fe deficiency and reduced serum ferritin levels lead to significant increases in Mn concentrations [157] in infants [153], children [158], and adults [123]. There is increasing evidence that the metabolism of metals and their interactions may differ between women and men. The epidemiological studies that have been conducted on metal levels and toxicity usually describe men and women separately, without considering specific periods in women’s lives. However, menstruation is known to cause Fe deficiency, which may be related to the increased gastrointestinal absorption of toxic metals [146,147]. In turn, Mn accumulation occurs during pregnancy due to the dysregulation of Fe metabolism [159]. Higher blood concentrations of Mn are observed in women of reproductive age than in men [123,152]. There is also a significant increase in mean Mn levels during pregnancy [128,160]. On the other hand, Mn levels decrease in menopausal women [126].

Changes in Mn levels at different stages of a woman’s life are related to fluctuations in Fe and ferritin levels. Unfortunately, it is important to remember that low Fe levels increase the risk of Mn accumulation in the brain [161].

Mn and Fe ions (Mn^2+^ and Fe^2+^) share the same routes of intracellular uptake and storage because they have similar coordination preferences and can compete for the same protein binding sites. In 2022 a paper was published [162] showing that excessive exposure to Mn disrupts the biosynthesis of coenzyme Q (CoQ), particularly the penultimate step, which is catalyzed by the mitochondrial diiron hydroxylase Coq7. CoQ deficiency disrupts oxidative phosphorylation in the mitochondria and is a cause of premature cell death. The authors of the study emphasize that the imbalance between the cellular Fe and Mn pools, which compete for the same compatible protein binding sites, is responsible for the toxicity of accumulated Mn. The disruption of CoQ biosynthesis has serious consequences, particularly for the brain, because this lipid, as an electron carrier and antioxidant, supports cellular processes such as fatty acid and pyrimidine metabolism. Mn overload leads to secondary CoQ deficiency. The correction of mitochondrial function can be achieved by increasing Coq7 protein levels or supplementation with the main group analogue of CoQ.

#### 2.6.3. Zinc (Zn), Copper (Cu), Lead (Pb), and Calcium (Ca)

Mn and Zn compete for absorption in the gut, and high intakes of one metal may interfere with absorption of the other. Animal studies have suggested that high intakes of Mn can lead to reduced Zn absorption and growth retardation [150,163]. Zn is known to play a protective role against Mn toxicity as it can compete with Mn for binding sites in enzymes and transport proteins.

Mn and Cu are both essential cofactors for several enzymes involved in antioxidant defense and energy metabolism. Studies have suggested that high intakes of Mn can interfere with Cu metabolism and lead to Cu deficiency and neurological symptoms in animals and humans [164,165]. Mn and Pb are both neurotoxic metals that can accumulate in the brain and cause cognitive and behavioral deficits. Studies have suggested that Mn may exacerbate the toxic effects of Pb and increase the risk of neurological damage in children [166,167].

Studies show a small effect of Ca supplementation on Mn metabolism in adults [143]. Ca supplementation in the form of calcium carbonate and calcium phosphate (500 mg/day) slightly decreased the bioavailability of Mn. The authors of the study emphasize that milk as a source of Ca had the least effect [168].

## 3. The Role of Mn in the Body

### 3.1. Mn-Dependent Enzymes

Mn is a cofactor for many key enzymes, i.e., Mn-superoxide dismutase (MnSOD), arginase, glutamine synthetase, glycosyltransferase and xylosyltransferase, pyruvate carboxylase, phosphoenolpyruvate carboxykinase, isocitrate dehydrogenase, the formation of ɑ-ketoglutarate (Krebs cycle), and serine/threonine phosphatase. Mn deficiency leads to a decrease in the activity of these enzymes. Mn is a cofactor for many enzymes, such as for instance glycosyl-, and xylozyl-transferases taking part in mucopolysaccharide biosynthesis. In several enzymes, Mn cannot be replaced by other metals. These are Mn-dependent enzymes, including arginase, agmatinase, glutamine synthetase, and MnSOD [83]. Mitochondrial superoxide dismutase and pyruvate carboxylase are two Mn-dependent metalloenzymes [146]. MnSOD is the major mitochondrial enzymatic antioxidant involved in the conversion of superoxide anion to hydrogen peroxide [169]. Arginase is involved in the urea cycle, which converts L-arginine to L-ornitine and urea. In addition, arginase activity in macrophages is associated with NO synthase and NO production, initiating a cascade of free radical formation [170]. Mn also activates other enzyme systems that are important for the synthesis of polysaccharides, glycoprotein, FA, and urea, such as oxidoreductases, namely dehydrogenases and lipoxygenase, which catalyze the oxidation of polyunsaturated fatty acids [171]. Mn is important for the production of phosphatidylinositol, which is used to biosynthesize the phosphatidylinositol 3-kinase family, which is involved in intracellular signaling. Mn deficiency also leads to a decrease in the activity of other enzymes, i.e., glycosyltransferases and xylosyltransferases, which are required for the synthesis of glycosaminoglycans in bone and cartilage, and prolidases and prolinases, which are involved in the metabolism of the extracellular matrix. Mn deficiency or genetic disorders that damage these enzymes adversely affect connective tissue function, angiogenesis, carcinogenicity, and mutagenesis [172]. Mn is also responsible for thyroid gland activity through deiodinase and is involved in the regulation of Mn homeostasis in the body.

### 3.2. The Role of Mn in Smooth Muscle Cell Contraction

Mn enters cardiomyocytes in the form of paramagnetic ions (Mn^2+^). Experiments using isolated perfused rat hearts have shown that myocardial slow Ca^2+^ channels are the major pathway for Mn^2+^ uptake and that Mn^2+^ acts as a pure Ca^2+^ competitor and a preferred substrate for slow Ca^2+^ channel entry [173].

The suppression of smooth muscle cell contraction by Mn has been linked to the alpha1 adrenergic receptor. Kalea et al. [174] investigated the effect of dietary Mn on the vascular contractile machinery in rat thoracic aortas. Rats were fed Mn-deficient or Mn-supplemented diets (<1 and 45–50 ppm Mn, respectively). It was observed that Mn-supplemented animals developed a lower maximal force of contraction and relaxation when contracted with the alpha(1)adrenergic receptor agonist L-phenylephrine compared to the Mn-deficient animals (*p* ≤ 0.001). In conclusion, the authors claimed that dietary Mn at levels of 45–50 ppm affects the contractile machinery by reducing maximal vascular contraction to alpha(1) adrenergic agonist. The observed vascular sensitivity was significantly greater in the Mn-deficient animals compared with the Mn-supplemented group (*p* ≤ 0.001).

### 3.3. Mn Transport

To reach the brain, Mn, like other essential metals, has to cross barriers such as the blood–brain and blood–cerebrospinal fluid, usually as Mn^2+^, Mn-citrate, Mn^3+^-transferrin, or alpha-2-macroglobulin, by various routes. Transport proteins are located in neurons, microglia, or astrocytes. The correct functioning of exporters and importers maintains intracellular Mn levels in the physiological range [175]. These transporters are usually not specific to Mn and can also transfer other metals.

Examples include transferrin receptors (TR), which are involved in the transport of Fe and Mn [176,177], and divalent metal ion transporter 1 (DMT1), which transports ferrous iron and other divalent metal ions [178,179]. Zn, Fe, and Mn with similar chemical properties compete for other common transport transmembrane proteins ZIP14 and ZIP8 [180]. Other examples of Mn transporters/importers include the citrate transporter, the dopamine transporter (DAT), and calcium channels [181]. DMT1 is highly expressed in several brain regions, including the caudate nucleus, putamen and substantia nigra [182], the subventricular zone of the brain, and the rostral migratory stream [183]. DMT1 is also known to be expressed in glial cells in the neocortex, subcortical white matter, and hippocampus [184]. Olfactory DMT1 is involved in Mn transport to the brain when exposed by inhalation [185]. ZIP8 plays an important role in Mn transport through the lungs and across the blood–cerebrospinal fluid (CSF) barrier, as it is highly expressed in the alveolar epithelium [186] and in the endothelial cells of cerebral capillaries [187]. ZIP14 in turn prevents the accumulation of Mn in the brain. Mutations in the SLC39A14 gene, which encodes ZIP14, cause Mn accumulation in the brain [188,189,190].

Another transmembrane protein is ferroportin (Fpn). It is expressed in the plasma membrane of various brain cells and is involved in the transport of Mn from the cell to the extracellular space. Thus, the overexpression of Fpn plays a protective role by reducing Mn accumulation [32,191]. SLC30A10 (ZNT10) mediates Mn efflux. SLC30A10 maintains intracellular Mn at physiological levels, thereby protecting cells from Mn toxicity. This transporter has been identified in Mn-induced parkinsonism [192,193,194,195]. In this disease, the retention of Mn is observed even in the absence of Mn exposure. Another Mn exporter is ATP13A2. Its overexpression reduces the intracellular concentration of Mn, while mutations are associated with the loss of function of this protein. Such ATP13A2 mutations have been identified in Lewy body disease [196].

In addition, many transporters (calcium uniporter, PARK9, and SPCA1) have been identified as being involved in the intracellular storage of Mn in different organelles such as lysosomes, Golgi apparatus, endosomes, mitochondria, and nucleus [197,198,199]. In terms of Mn homeostasis, TMEM165 is a major contributor to Mn homeostasis in the Golgi compartments, which is the site of terminal glycosylation. The regulation of TMEM165 is affected by cytosolic Mn concentration. Mn homeostasis in cells has been studied in yeast [99,200] and *C. elegans* [201]. Mn transporters have been found to reduce the influx of Mn into the cell under increased stress or to increase it under Mn deficiency. A schematic representation of the imbalance in the Mn transport system is shown in Figure 4.

### 3.4. Oxidative Stress/Inflammation

The brain, particularly the basal ganglia, including the globus pallidus and striatum, is known to be highly susceptible to oxidative damage [202,203]. Therefore, antioxidant barriers are important to maintain its structure and physiological function. Oxidative stress is the state of imbalance between reactive oxygen and nitrogen species, such as superoxide radical (O_2_^--^), hydrogen peroxide (H_2_O_2_), and hydroxyl radical (-OH), and neutralization by endogenous antioxidant defense mechanisms [204]. The bulk of the antioxidant defense system consists of antioxidant enzymes, including isoforms of superoxide dismutase (SOD), catalase (CAT), and glutathione peroxidase (GPx) [205,206].

Mn exposure has been shown to be responsible for free radical formation and activation of nitric oxide synthase in astrocytes [207], which stimulates the production of pro-inflammatory cytokines and chemokines (including TNF-α, IL-1, IL-6, interferons, CCL2, and CCL5) by activating the NF-κB pathway [207,208,209,210,211,212,213].

The weakening of the antioxidant defense system, mainly by inhibiting glutathione synthesis [214], exposes the brain to the neurotoxic effects of free radicals formed as a result of dopamine oxidation [215,216]. The induction of a state of oxidative stress is associated with the activation of pro-apoptotic processes and a range of cellular damage in both primary neurons and astrocytes [217,218]. The cell membrane and DNA are disrupted by the oxidation of polyunsaturated fatty acids [219] and nucleic acids [220]. Microglial cells and astrocytes are particularly susceptible to inflammation due to the expression of DMTI and TR transporters [221,222,223], which are expressed on the surface of astrocytes. Glial neuroinflammation occurs in patients exposed to Mn [224,225,226,227,228], mainly in the basal ganglia [78].

Filipov et al. [212] confirm the increase in IL-6 and TNF-α production induced by exposure to Mn. A study on HAPI microglial cell lines has shown that Mn causes an increase in ERK and p38 activation, which enhances cytokine production [228,229], and an increase in the expression of the pro-inflammatory genes nitric oxide synthase 2 (Nos2), IL-1β, and caspase 1 [230]. Popichak et al. also confirmed the increased expression of the inflammatory genes Nos2, TNF-α, Cc15, IL-6, Ccr2, and IL-1βw in cultured astrocytes and microglia exposed to Mn [209]. Harischandra et al. described the release of pro-inflammatory cytokines (IL-12, IL-1b, and IL-6) and α-Syn-containing exosomes as well as an increase in Iba-1 and iNOS expression by Mn-exposed microglia [231]. A report by Chen et al. demonstrated microglial neuroinflammation after Mn exposure and autophagy dysfunction in vivo and in vitro [232]. A report by Sarkar et al. showed an increase in NLRP3 and the maturation of the inflammatory cytokine IL-1β [233].

In addition, endoplasmic reticulum (ER) stress has been strongly implicated in Mn-induced toxicity. However, reports of associations between Mn exposure and ER stress and neurological disorders are limited. Mn exposure leads to the upregulation of ER-stress-related genes, including the FK506 binding protein family, and has been shown to be toxic in *Caenorhabditis elegans* [234]. Mn promotes ER stress and ER-stress-mediated apoptosis through the activation of the caspase family in rat striatum and dopaminergic neurons, resulting in a parkinsonian phenotype associated with movement disorders [235,236].

### 3.5. Neurotransmitters

Trace metals present in presynaptic vesicles are released into the synaptic cleft. It has therefore been suggested that metals such as Zn and Mn may affect the release of neurotransmitters into the synaptic cleft by acting on ion channels and neurotransmitter receptors and transporters [83]. Mn has been shown to disrupt the function of the neurotransmitters dopamine (DA), glutamate, gamma-aminobutyric acid (GABA), and acetylcholine (ACh) [57,58,59,60,61,62]. Disturbances in neurotransmission lead to motor, cognitive, and behavioral disorders.

In animal models, Mn accumulation has been shown to reduce DA levels in substantia nigra pars compacta neurons in the basal ganglia [237,238,239,240] and in the striatum [241]. The weakening of dopaminergic neurotransmission and damage to dopaminergic neurons due to exposure to Mn has also been confirmed in humans [242]. The key role is played by the reduction in the expression of tyrosine hydroxylase (TH) [243,244,245] and the transcription factor repressor element-1 silencing transcription factor (REST) in dopaminergic cells by Mn [244].

Mn also affects glutamate neurotransmission in the brain by impairing its transporters, i.e., aspartate 1 (GLAST) and glutamate transporter 1 (GLT-1), and the N-methyl-D-aspartate (NMDA) receptor [246,247]. As a result of reduced transporter expression, glutamate accumulates outside the cell [248,249], and postsynaptic receptors are stimulated [250]. Ultimately, exposure to Mn leads to glutamate-induced excitotoxic neuronal damage [251].

Mn also affects the cholinergic system, resulting in impaired intellectual performance, particularly in young brains [252]. Mn has been shown to inhibit acetylcholinesterase (AChE) [253] and choline acetyltransferase (ChAT) [254]. The result is an accumulation of Ca and a reduction in the amount of choline in striatal cholinergic terminals. Mn impairs choline transport in the hippocampus, cerebral cortex, and basal ganglia [255], and ACh-binding in the prefrontal cortex [256].

The effect of Mn on GABAergic neurotransmission has also been observed. Exposure to Mn reduced brain gamma-aminobutyric acid (GABA) levels, lowered seizure thresholds, and increased seizure duration in rat studies [257]. Similar results were found in another study, i.e., a decrease in the rate of accumulation of L-glutamate and gamma-aminobutyrate (GABA) and decreased (3)H-GABA uptake in synaptosomes in rat forebrain nerve terminals due to dietary Mn overload [258]. At the same time, it has been shown that Mn exposure causes an increase in extracellular GABA concentration in the striatum through the altered expression of transport and receptor proteins, e.g., Mn exposure alters extracellular GABA, GABA receptor and transporter protein, and mRNA levels in developing rat brains [259]. In humans, exposure to Mn showed increased GABA levels in the thalamus and adjacent brain areas but not in the globus pallidus, which had the most Mn deposits [260]. The authors of the above study emphasize that the effect of Mn on the expression of GAT-1 mRNA as well as GABA(A) and GABA(B) proteins is region-dependent. While Mn exposure decreased GAT-1 protein expression by about 50% in the substantia nigra while increasing mRNA expression about fourfold, mRNA expression was reduced in the caudate putamen without affecting protein expression.

### 3.6. Protein Aggregation

The SNCA-encoded protein α-synuclein (αSyn) plays an important role in Mn-induced neurotoxicity. Mn increases the expression of αSyn and induces conformational changes that facilitate its aggregation [261]. αSyn aggregation is characteristic of some neurodegenerative diseases, such as PD, which is characterized by Lewy bodies with αSyn in the brain [262]. Mn is responsible not only for the misfolding of αSyn, but also for its prion-like transfer from cell to cell [231]. However, it should be emphasized that short-term exposure to Mn and the associated overexpression of αSyn has a neuroprotective effect on dopaminergic cells [263], whereas chronic exposure is clearly neurotoxic. Recent studies have confirmed the role of Mn in packaging αSyn into exosomes that are released into the extracellular environment [264]. Studies in humans with chronic occupational exposure to Mn confirmed the presence of misfolded αSyn in serum exosomes. The stereotaxic delivery of αSyn-containing exosomes, isolated from Mn^2+^-treated αSyn-expressing cells, into the striatum induced parkinsonian-like pathological features in mice.

Mn is an important pathogenetic factor in Alzheimer’s disease (AD). This is confirmed by studies, including a 2014 study [265] conducted in a group of 40 older adults that measured Mn levels in whole blood and plasma amyloid-β (Aβ) peptides. A correlation between Mn levels and the cognitive status of the participants was confirmed. In addition, Aβ peptide levels increased with increasing plasma Mn levels. Not only aggregated amyloid-β (Aβ) peptides, but also elevated levels of Mn have been found in the brains of AD’s patients. The content of Mn, together with Si, Sn, and Al, was significantly higher in the parietal cortex of AD brains compared to controls [266].

There are several mechanisms of Mn-induced plaque formation or amyloidogenesis characteristic of AD brain pathology. Mn exposure may cause a reduction in cholinergic neurons in the basal region of the BF forebrain by increasing the aggregation and levels of amyloid beta (Aβ) and phosphorylated tau protein (pTau), altering the expression of heat shock proteins (HSP90 and HSP70), thus disrupting P20S proteasome activity and generating oxidative stress [267].

Gene expression studies in the frontal cortex of primates (*Cynomolgus macaques*) exposed to Mn showed changes in the expression of several genes, but Aβ precursor-like protein (Aβ) 1 (APLP1) was the most regulated gene. The aggregation of αSyn in grey and white matter and the presence of diffuse A-beta plaques have been observed after exposure to Mn [268].

The work of Lin et al. [269] described the mechanism of induction of atherosclerotic plaques by Mn using the transgenic mouse model (3 × Tg-AD) and mouse-derived microglia and neuroblastoma cell lines. Increased Aβ expression and Aβ plaques were found in the cerebral cortex and hippocampus. Mn exposure was also shown to be amyloidogenic by increasing Aβ production through reduced α-secretase cleavage activity.

Mn increased the release of inflammatory cytokines, namely interleukin-1β (IL-1β) and tumor necrosis factor-α (TNF-α), from microglia, which stimulated BACE1 gene and protein expression, and consequently Aβ production. However, Mn alone did not affect amyloid-beta precursor protein (APP) and beta secretase (BACE1) expression nor Aβ production in N2a cells unless the cells were co-cultured with microglial-conditioned media.

The kinetics of Mn-mediated amyloid fibrillation, exemplified by egg-white lysozyme, has been elucidated at the molecular level by Raman spectroscopy, atomic force microscopy (AFM), thioflavin T (ThT) fluorescence, and UV-vis absorption spectroscopy assays [270]. The unfolding of tertiary protein structures has been shown to be effectively accelerated by Mn^2+^ ions, which tend to form amorphous aggregates rather than amyloid fibrils. Mn^2+^ is a specific accelerator of the secondary structure transition from α- to β-helix.

## 4. Brain Function and Neurodevelopment

Mn is essential to support various brain functions and development, and to reduce oxidative stress [271,272]. Deficiency of this micronutrient can lead to cognitive deficits [273]. Mn is involved in the activation of several enzymes, including arginase, glutamine synthetase (GS), pyruvate carboxylase, and Mn superoxide dismutase (Mn-SOD). The enzymes arginase and agmatinase, which are part of the urea cycle in the brain, require Mn for their function [41]. Changes in Mn levels have been linked to changes in the urea cycle, insulin signaling, and cellular autophagy processes in the brain [194,242,274,275,276]. Mn is a cofactor for glutamine synthetase in astrocytes, which converts glutamate to glutamine. Therefore, an increase in glutamine content in the hypothalamus [83,277] and increased the production of serotonin and decreased secretion of dopamine and neuropeptide Y have been observed in association with Mn accumulation. Low levels of Mn are associated with reduced glutamine synthetase activity in the brain, glutamate accumulation, increased convulsive brain activity, and the development of epilepsy [278].

Studies have shown that after crossing the blood–brain barrier, Mn accumulates mainly in the hypothalamus or pituitary gland [279]. However, the affinity of Mn for different brain areas varies. In our previous study [84], the affinity of different brain areas for Mn can be ordered as follows: insula > hippocampus > precentral gyrus ~ head of caudate nucleus. Besides the brain, Mn accumulates in several mitochondria-rich organs, e.g., pancreas and liver [280,281].

Elevated Mn levels are usually the result of overexposure to the anthropogenic environment. The association between environmental Mn exposure and neurodegeneration and cerebrovascular lesions has been demonstrated. One of these is an epidemiological study of a group of 936 adults (442 men and 494 women) from the Republic of Korea [282]. Parameters such as regional cortical thickness and white matter hyperintensity volume (an indicator of cerebrovascular changes) were assessed using 3 T brain MRI images. A significant correlation was found between ambient Mn concentration and thinner parietal cortex (β = −0.02 mm; 95% confidence interval (CI), −0.05 to −0.01) and occipital cortex (β = −0.03 mm; 95% CI, −0.04 to −0.01) in men but not in women. An increase in ambient Mn concentration was also associated with a larger volume of deep white matter hyperintensities in men (β = 772.4 mm^3^; 95% CI, 36.9 to 1508.0). Mn is a risk factor for many neurological diseases. Excess Mn can affect neurophysiological processes, such as motor function and higher-order cognitive functions or learning deficits [283]. The neurotoxicity of Mn results from, among other things, the disruption of dopaminergic function. This is clinically manifested by Parkinson’s-like movement disorders, known as manganism [284,285,286].

The molecular mechanisms of Mn neurotoxicity and transport mechanisms are currently under intense investigation. Transcriptional response pathways to Mn overexposure have been investigated in the *Caenorhabditis elegans* (*C. elegans*) model using RNA-Seq analysis, and bioavailability and survival assays [287]. A large number of differentially expressed genes (DEGs) were identified and oxidative nucleotide damage, the unfolded protein response, and innate immunity were suggested as the main pathways of the Mn exposure damage response. Recently, a comprehensive review of molecular targets of Mn-induced neurotoxicity has been published [288].

### 4.1. Neurodegenerative Disorders

Mn-induced neurotoxicity at the cellular level is mainly due to impaired mitochondrial function, impaired neurotransmitter metabolism, effects on Fe homeostasis and induced oxidative stress leading to impaired glial cell function [218,289,290,291,292]. Mn has been shown to affect microglia and astrocytes, regulating the activation of pro-inflammatory responses [293]. Neurodegenerative diseases, such as Parkinson’s disease (PD), Alzheimer’s disease (AD), Huntington’s disease (HD), and amyotrophic lateral sclerosis (ALS) are characterized by metal dyshomeostasis, which may contribute to disease-specific protein aggregations, such as αSyn [293,294,295].

While the neuropsychological effects of high levels of Mn exposure are well known, the effects of lower levels of exposure are less well understood. A 2011 study [296] investigated the effects of lower levels of occupational exposure to Mn (median = 12.9 μg/m^3^) in adult men. Cumulative Mn exposure indices (Mn-CEIs) were correlated with performance on cognitive, motor control, and psychological tests. We found that an increase in Mn-CEI was significantly associated with slower reaction time on a continuous performance test (CPT; *p* < 0.01), as well as worse mood on several scales of the mood state profile (POMS; confused, tired, and composed of tired and energetic, all *p* ≤ 0.03) and a decrease in fine motor control (*p* = 0.04).

Children are more susceptible than adults to the adverse effects of Mn exposure because their nervous systems and brains are still developing. Excessive exposure to Mn in children can lead to nervous system damage, behavioral disorders, memory problems, learning disabilities, and growth and developmental retardation. Exposure to Mn in water has been shown to be associated with poorer neurobehavioral outcomes in children, even at the low levels commonly found in North America [297]. In addition, Mn exposure and low blood Pb levels were significantly associated with poor postural balance in children living near a ferromanganese refinery [298]. Some studies on the association between perinatal and childhood Mn exposure and neurodevelopmental disorders are summarized in Table 2.

### 4.2. Autism Spectrum Disorder (ASD)

Evidence suggests an association between exposure to certain elements [316], including Mn during pregnancy, and attention deficit hyperactivity disorder (ADHD) or autism spectrum disorder (ASD) in children [317]. Many studies have linked ASD to a state of oxidative stress [318,319,320], which induces autoimmunity in children with ASD [321,322]. This belief is linked to the results of studies that have confirmed lower levels of glutathione (GSH) and the ratio of reduced to oxidized GSH in children with ASD compared to children without ASD [323,324]. Changes in other markers of oxidative stress, such as increased lipid peroxidation [325,326] and altered platelet dense granule morphology [327], blood–brain barrier (BBB) deficits, and a disruption of neurovascular signaling and of the BBB with neuroinflammation have also been observed in the ASD group [328]. In addition to xenobiotics, heavy metals, such as Pb, Hg, arsenic (As), and Cd, are also responsible for the oxidative stress state. It has been confirmed that elevated blood levels of Pb [329] and hair aluminum (Al) levels were significantly elevated by 29% in children with ASD [330]. Trace element imbalances and the pathogenesis and severity of autistic symptoms have been discussed in original and review papers [331,332,333].

The key role in protecting against oxidative stress is played by the cellular antioxidant glutathione (GSH), which binds heavy metals.

In 2020, the study on the levels of several toxic elements (i.e., Pb, Hg, As, Cd, and Mn) and the risk of developing ASD was conducted [334]. While no association was reported between ASD status and these elements, a significant interaction was found between the glutathione S-transferase (GST) family gene GSTP1 and blood Mn concentration (BMC). Among children with many genotypes for GSTP1, those with BMC ≥ 12 μg L^−1^ were 4.6 and 4.27 times more likely to develop ASD compared to those with BMC < 12 μg L^−1^. The findings were confirmed in 2018 in a larger population of 163 case–control pairs of children aged 2–8 years from Jamaica [334,335,336].

As a micronutrient necessary for brain development, Mn must be ingested with food, which is also the main source of exposure. High levels of Mn can pose a significant threat to fetuses, infants and young children because of the ability of Mn to cross the placenta and the fact that children can absorb more Mn than adults [337]. Epidemiological studies evaluating the association between prenatal or postnatal Mn exposure and neuropsychological development are inconsistent. The relationship between Mn concentrations in tissue samples and neurodevelopment is an inverted U-shape. Thus, cognitive and behavioral problems occur at the highest and lowest exposure levels [302].

To assess prenatal exposure to Mn, different matrices are commonly used for testing, such as samples from newborn hair, whole blood, deciduous child teeth, or umbilical cord blood. Blood is by far the most commonly used to assess exposure to Mn [338]. However, it should be noted that its half-life in blood is less than 30 days, so the result obtained may only be indicative of recent exposure [103]. The measurement of hair Mn levels may be affected by contamination [339]. Neurodevelopmental disorders are assessed by children’s scores on the Mental Development Index (MDI) and Psychomotor Development Index (PDI) of the Bayley Scales of Infant Development. Several studies have attempted to determine the relationship between prenatal exposure to Mn and neurodevelopment [37,38,303,340,341,342]. Some studies have shown that excessive exposure to Mn can be toxic and cause neurological symptoms, including behavioral changes [343,344,345,346]. Some research has even found a possible link between exposure to high levels of Mn and ASD.

Many epidemiological studies describe higher levels of Mn in various tissues, e.g., hair, blood [345,347,348,349,350], and umbilical cord blood [301]. The effects of exposure are usually associated with lower IQ, poorer learning and memory skills, and poorer motor and manual coordination. Similar relationships have been observed between maternal Mn levels and mental and psychomotor development in young children [340]. It should be emphasized that Mn concentrations in cord blood samples are usually higher than in maternal blood [351]. In contrast, some researchers have reported lower levels of Mn in the hair and serum/plasma of children with ASD compared to children without ASD [83,156,331,352,353,354,355,356,357].

### 4.3. Attention Deficit Hyperactivity Disorder (ADHD)

ADHD is a disorder that affects about 7% of children. It is characterized by inattention, impulsivity, and hyperactivity [358]. In the review by Tarnacka et al. [359], the authors describe the association between neurological disorders and excess Cu, Fe, and Mn. Many authors report an increase in the incidence of ADHD with increasing blood levels of Mn. Most researchers agree that exposure to Mn in early childhood or prenatally predisposes the child to future susceptibility to attention deficits. The role of Mn in the etiology of ADHD is not fully understood, although its involvement in striatal dopamine neurotransmission is thought to play some role [360].

The treatment of ADHD with methylphenidate has been shown to reduce blood levels of Mn [361]. This has been confirmed in rat studies [362]. The involvement of Mn in the pathogenesis of ADHD is still controversial. There are also other reports, such as that of Lee et al. [363], which investigated the relationships between urinary levels of Mn, Pb, Cd, Hg, antimony (Sb), and Bi and ADHD symptoms. The study included almost 120 patients with an average age of 8.1 years with ADHD inattentive type, ADHD hyperactive/impulsive type, and control children. However, the authors did not find any significant differences in Mn between the groups. The measured urinary Mn concentration in the controls (>2 µg L^−1^) was even higher than in the studied groups (<2 µg L^−1^).

### 4.4. Manganism

Mn accumulates in the body as a result of the excessive exposure or disruption of its homeostasis due to liver diseases, such as hepatobiliary insufficiency and cholestasis. The accumulation of Mn in the brain causes a group of symptoms called manganism [78]. Mn has been shown to accumulate in the subcortical structures of the basal ganglia, particularly the substantia nigra pars compacta, pallidum, and striatum [151,364,365]. The accumulation of Mn in the brain affects the hypothalamic–pituitary system, resulting in increased prolactin levels and decreased dopamine production [287]. Excess manganese mediates the oxidation of DOPA to melanin. Therefore, Mn dyshomeostasis affects skin pigmentation [366,367].

Excess Mn is associated with neurotoxic effects [368]. Severe dysfunction occurs at both high and low levels of lifetime exposure [369]. Early clinical manifestations of manganism include weakness, headache, decreased appetite, sleep disturbances, muscle excitability, and clumsiness. Later, speech problems, difficulty walking, and increased reflexes are observed. In a full-blown PD-like syndrome, symptoms appear, such as emotional instability, hallucinations, manic or depressive psychosis, muscular dystonia, tremors, and rigidity [370]. There are many similarities between the symptoms of manganism and idiopathic Parkinson’s disease (PD) [371]. Unlike PD, manganism does not produce Lewy bodies. In addition, manganism is also characterized by the presence of dystonia and the absence of resting tremor.

#### 4.4.1. Parkinson Disease (PD)

There is a well-established association between occupational exposure to Mn and the risk of developing Parkinson’s disease, which has been confirmed by several studies in humans and animal models [372,373,374,375,376,377,378]. Epidemiological studies have reported positive associations between PD and long-term exposure to Mn, as well as other metals such as Hg, Cu, Pb, Zn, Fe, and Al [379]. Although the exact mechanism by which Mn causes PD is not fully understood, it is thought to involve oxidative stress, inflammation, and damage to dopaminergic neurons in the substantia nigra [380], although other brain regions may also be involved [295].

Metabolic abnormalities and the accumulation of cytotoxic glutamate are also seen in both PD and manganism [353]. Although the symptoms of manganism are similar to those of Parkinson’s disease, the dysfunctions involve different areas of the brain. In manganism, Mn accumulation occurs mainly in the pallidum—the GABA-rich area [77]. In PD, changes are observed in the dopaminergic nuclei of the substantia nigra. It is therefore understandable that L-DOPA therapy is the most effective in PD [167]; however, the research is not consistent [381,382]. Some reports describe no changes in GABA in the striatum and thalamus after Mn exposure and in PD patients [383]. Mn-induced parkinsonism is associated with dopaminergic (DAergic) neurodegeneration [384]. The cause of neurodegeneration is thought to be the upregulation of divalent metal transporter 1 (DMT1) in the basal ganglia, which is associated with the loss of dopaminergic cells [385]. The altered function of dopaminergic neurons and Mn-induced changes in the dorsal striatum have been demonstrated by positron emission tomography (PET), single photon emission tomography (SPECT), and magnetic resonance imaging (MRI) studies [386,387].

Cell culture studies also confirm that DA neurons are sensitive to exposure to Mn [198]. Other investigators have observed increased levels of GABA in the thalamus, either in controls or in PD patients [381].

A key player in PD pathology appears to be αSyn, a protein with a natural tendency to aggregate into oligomers [190]. Mn has been shown to affect the function of the protein αSyn, which is found in Lewy bodies in the brains of PD patients [388].

#### 4.4.2. Alzheimer’s Disease (AD)

AD is most common in the elderly and is characterized by a state of mental dementia. The clinical picture of the disease is characterized by the presence of amyloid beta protein (Aβ) plaques in the brain and neurofibrillary tangles (NFT) derived from hyperphosphorylated tau protein in the neurocytes. Li et al. [389] presented current opinions on the influence of biometal dyshomeostasis and toxic metal accumulations in the development of AD in their review. In experimental studies, attention was paid to the use of vanadium (V) compounds in alleviating the effects of AD [390]. There is evidence that the abnormal accumulation of metals, like Fe, Cu, and Zn, causes an oxidative stress state resulting in cellular function damage responsible for the development of AD as well as other neurodegenerative diseases [391,392]. The disturbance of Mn homeostasis has been confirmed in many neurodegenerative diseases, i.e., AD, PD, amyotrophic lateral sclerosis, and prion disease [284,393]. That is why environmental exposure to higher doses of Mn may promote the development of neurodegeneration [394]. Mn neurotoxicity is associated among others, with the inhibition of the electron transport chain in brain mitochondria: decreasing ATP formation, enhancing cellular oxidative stress by ROS generation [395,396,397,398,399], and the loss of membrane potential [58,59]. Mn accumulation in the brain is associated with AD possibly through the overproduction and aggregation of amyloid-β (Aβ) found in AD patients.

A systematic review and meta-analysis of 2000 people suffering from AD performed recently by Du et al. showed decreased serum Mn levels of AD and patients with mild cognitive impairment (MCI) [400]. Patients with AD had lower serum Mn levels, suggesting that Mn deficiency may be a risk factor for AD [400]. The expression level of MnSOD in the hippocampal CA1–CA4 region of AD patients was 3–11-fold higher than that of the control group, suggesting that the normal compensatory mechanism of Mn-dependent antioxidant enzymes may not be sufficient to protect the hippocampus from free radical oxidative damage [401,402]. This may also be explained by the influence of Mn on glutamate accumulation because of the inactivation of the Mn-dependence enzyme glutamine synthetase (GS). Oppositely, Tong et al. [265] reported increases in Mn in plasma associated with increased Aβ peptides. Balmus et al. [403] demonstrated that high Mn levels in AD patients accompanied by decreased glutathione levels enhanced MDA (a measure of lipid peroxidation) as well as levels of protein carbonyls (a measure of protein oxidation). Another work showed the association of Mn accumulation with diffuse Aβ plaques and tau, confirming a direct impact of Mn on the development of AD [404]. Intravenous Mn (3.3–5.0 mg/kg) in Cynomologous Macaques increased brain Mn, upregulated the expression of amyloid-β-like protein 1 mRNA, and led to diffuse Aβ plaques and degenerating cells. These changes are accompanied by impaired spatial working memory and fine motor skills, and increased compulsive-like behaviors [405]. Alpha-synuclein (αSyn) is a chaperon protein that predominantly localizes to the presynaptic terminal. It can play important roles in the regulation of synaptic plasticity, vesicle transport, and dopaminergic neurotransmission [406]. αSyn has a poor affinity for Mn^2+^ in its C-terminal binding site, however Mn^2+^ can trigger the misfolding and accumulation of αSyn protein [263,356,388,407,408]. In AD patients, the loss of neurons in the basal forebrain, changes in the cholinergic system in the cerebral cortex and hippocampus occur, which manifests in the form of locomotor, emotional, behavioral and cognitive dysfunctions [409,410]. Studies on rats have shown that even the short-term administration of Mn causes an increase in AChE activity in the hypothalamus, cerebellum, and striatum [411], which increases Aβ formation within and around amyloid plaques. In turn, chronic exposure to Mn in drinking water (40 mg Mn/kg daily for 2 years) resulted in a decrease in AChE activity and an increase in the pro-inflammatory marker F2-isoprostane [253]. Exposure to Mn via intranasal administration resulted in increased oxidative stress and decreased levels of nicotinic acetylcholine receptors in the prefrontal cortex, which is critical for cognition [256]. Despite AD pathogenesis being associated with an aggregation of amyloid-β (Aβ) and the hyperphosphorylated tau protein [379], recent spectroscopy and fluorescence experiments showed that Mn does not change aggregation kinetics and the fibril morphology of Aβ [412,413]. A lack of consistency between a few reports presenting the importance of Mn in AD development may be a consequence of, for instance, additional health circumstances, Mn doses, its bioavailability, or chemical form [402].

By screening differentially expressed genes (DEGs) that are crucial for the development of Mn-induced AD using Venn analysis, gene function (GO) analysis, and KEGG pathway analysis, Ling et al. selected 140 upregulated genes and 267 down-regulated genes [414], the most important of which are INSR, VEGFA, PRKACB, DLG4, and BCL2. KEGG pathway analysis identified the top five molecular mechanisms of Mn-induced neurotoxicity, which included cytokine–cytokine receptor interaction, apoptosis, oxidative phosphorylation, the Toll-like receptor signaling pathway, and the insulin signaling pathway. The authors of the study selected one of the tyrosine phosphorylation inhibitors (tyrphostin AG-825) as a drug capable of inhibiting AD caused by Mn neurotoxicity.

#### 4.4.3. Huntington’s Disease (HD)

HD is an autosomal dominant neurodegenerative disease manifesting a progressive breakdown of nerve cells, mainly the medium spiny neurons of the corpus striatum in the brain, leading to a decline in motor function, cognition, and psychiatric symptoms. This is a genetic disorder accompanied by the impaired activity of Mn-related enzyme glutamine synthetase, which converts glutamate to glutamine, in the striatum. The mutant protein huntingtin results from the expansion of a glutamine-encoding CAG triplet repeat in the Huntingtin (HTT) gene, which causes HD [415]. HD is also characterized by decreased arginase and SOD2 activity. The pathophysiological mechanisms of HD are poorly understood. The studies use transgenic animal models provide insight into potential causes factors. There are many pieces of experimental evidence suggesting the participation of excitotoxicity and altered calcium signaling, mitochondrial dysfunction, oxidative stress, protein aggregation, and others [416,417,418]. The HTT gene has also been functionally linked to iron (Fe) metabolism, and HD patients show alterations in brain and peripheral Fe homeostasis. It is known that many neurodegenerative disorders have been associated with altered metal homeostasis [419]. Some studies revealed reduced net Mn uptake specifically into the striatum following systemic Mn exposure of a mouse model of HD [420]. Williams et al. suggested that the cytotoxicity mechanisms of Mn is likely similar in wild-type and mutant HTT cells because they exhibit similar increases in cellular Mn-burden cell death after exposure. Trettel et al. [421] compared a striatal neuronal cell line model of HD and a wild-type cells. They found alterations in Fe signaling and an increased expression of the TfR protein. This observation was confirmed by Bartzokis et al. [422] who noticed that HD patients display elevated levels of Fe in the corpus striatum. It was proven that changes in Fe homeostasis can alter Mn homeostasis [32,423] because the transferrin receptor (TfR) as well as transports, like DMT1 or ferroportin, have an affinity for both biometals. Williams et al. [295] revealed that Fe(III) exposure decreased Mn neurotoxicity equally for wild-type and mutant STHdhQ111/Q111 striatal cell line. Western blot analysis showed that TfR levels are reduced in the mutant cell line, whereas levels of DMT1 is unchanged. Recent studies have found that genetic variations associated with HD may increase susceptibility to Mn toxicity [358,424]. There have been studies investigating the potential link between HD and Mn exposure. While the exact relationship between HD and Mn exposure is still unclear, there is evidence to suggest that excessive Mn exposure may worsen the symptoms of HD. Many researchers hypothesize that iron dyshomeostasis induced by ferritin and ferroportin receptors plays a key role. Along with the impairment of Fe exchange, the export of Mn is also impaired, which leads to the accumulation of the latter in the CNS [393]. HD patients also have higher levels of Mn in their blood than healthy individuals [61]. The accumulation of Mn in brain tissue and the resulting degeneration of the basal ganglia in HD was reported. There is evidence of reduced Mn bioavailability in HD. In mice, exposure to Mn in the HD model has been observed to inhibit many Mn-dependent biological processes [425]. Despite the proven neurotoxicity of overexposure to Mn, maintaining adequate Mn levels is important to preserve its neurophysiological function. It appears that therapeutic exposure to Mn can alleviate certain neurodegenerative disorders in which Mn-dependent or -responsive processes are suppressed or deficient. It is worth noting that some HD phenotypes are corrected by Mn exposure. For example, an increase in the concentration of urea cycle metabolites in the striatum in a mouse model of HD before the onset of symptoms can be corrected by 1-week exposure to Mn. However, the molecular mechanisms by which HD affects Mn homeostasis are still under investigation.

### 4.5. Epilepsy

Mn is an essential trace element for the physiological function of the central nervous system. The first published report relating Mn and epilepsy was written by Griffiths and Fox in 1938 [426]. Alterations in Mn concentrations, either excessive or deficient, can be accompanied by convulsions. Mn-deficient rats have been shown to be more prone to seizures compared to Mn-sufficient rats. In addition, rats genetically susceptible to epilepsy have lower than normal concentrations of Mn in the brain and blood. Some epileptic patients have lower whole-blood Mn concentrations than non-epileptic individuals [427]. One study found that the blood levels of Mn in people with epilepsy of unknown origin were lower than in people whose epilepsy was caused by trauma (e.g., head injury) or disease [428], suggesting a possible genetic link between epilepsy and abnormal metabolism Mn [427,429].

### 4.6. Prion Diseases

The most common prion disease in humans is Creutzfeldt–Jakob disease. People with this disease have been shown to have higher levels of Mn in their blood and brain [104]. The role of Mn in the onset and development of this disease is not fully understood. However, it is thought that the interaction of Mn and other divalent metal ions with the PrPc prion protein may play a role, causing its misfolding and the accumulation of PrPSc isoforms [430,431].

### 4.7. Leigh-Like Syndrome

Leigh-like syndrome is neurological disorder characterized by a progressive psychomotor regression. It is characterized by low serum and high urine concentrations of Mn [432]. Leigh syndrome diseases, similarly to congenital disorder of glycosylation type II (CDG type II), are associated with hSLC39A8 mutations. The patients exhibit severe manganese (Mn) deficiency because hSLC39A8 mediates Mn uptake by the cells. The expression of hSLC39A8 mutants reduces mitochondrial Mn levels and activity of Mn-dependent mitochondrial superoxide dismutase MnSOD, and in finally increases oxidative stress.

## 5. Infectious Diseases

While deficiencies of such transition metals, namely Fe and Zn [433,434], are known to increase the incidence of infectious diseases in humans [435,436,437], there was no evidence that the amount of Mn(II) is associated with the occurrence or severity of an infectious disease. In recent years, however, it has been confirmed that the invasive microorganism uses Mn to adapt to the human host, mainly to mitigate the effects of oxidative stress [438]. Nutrient and mineral uptake by pathogenic bacteria is known to be necessary to maintain virulence in the environment of an infected host. It concerns especially Mn-centric organisms [439,440], including the lactic acid bacteria *Streptococcus pneumoniae* and *Lactobacillus plantarum*, the pathogen *Neisseria meningitidis*, and the UV-resistant *Deinococcus radiodurans*. Mn affects the viability and virulence of both Gram-positive and Gram-negative bacterial pathogens [441,442,443]. A lack of Mn worsens the virulence of pathogens due to the attack of free radicals, e.g., O^2−^ and H_2_O_2_ O^2−^, and the formation of the hydroxyl radical, OH^•^, in the Fenton reaction. Thus, the host defense mechanism covers micronutrient sequestration [444], e.g., through the extracellular action of calprotectin (CP) [444] belonging to the S100 family protein, which is found in neutrophils. In addition, resistance-associated macrophage protein 1 (NRAMP1) and related H^+^-linked transporters sequester both Mn(II) and Fe(II) from intracellular compartments against invading pathogen [434,445]. This action eliminates the activity of Mn(II) as an antioxidant being a cofactor for Mn superoxide dismutase (MnSOD). Past infections affect the status of trace elements. In 2022, a study was conducted that included 260 unvaccinated residents of the Mangistau region of the Caspian Sea region who recovered from COVID-19 (aged 18–60) [130]. The obtained results indicate that the content of Mn (0.642 (95% CI: 00.518; 0.795)), among others, in the hair of people who underwent a coronavirus infection was lower than in people who did not have this infection.

## 6. Genetic Diseases Associated with Mn Imbalance

### 6.1. Mn Deficiency

Since *SLC39A8* gene encodes the ZIP8 protein, which is a divalent metal ion transporter with a high affinity for Mn, patients with the *SLC39A8* mutation were Mn-deficient, resulting in defects in virtually every organ, tissue, and cell type, including bone defects, neurological defects, impaired fertility, and impaired glucose, lipid, and carbohydrate metabolism [446,447]. Defects currently associated with *SLC39A8* deficiency variants include Mn^2+^-deficient hypoglycosylation, among many other clinical disorders. Nebert et al. [448] reviewed many critical cellular functions, and clinical disorders in the case of deficiency of the ZIP8 transporter, starting with the early mouse studies that began in 1919. Lin et al. [449] performed animal model studies using *Slc39a8*-inducible global knockout mice, liver-specific knockout mice, and mice overexpressing *SLC39A8* in the liver. They found that the ZIP8 transporter is indeed involved in the recovery of Mn from bile and thus influences the activity of Mn-dependent enzymes, mainly arginase and β-1,4-galactosyltransferase.

Genetic defects in enzymes requiring Mn for catalytic activity manifest clinically as Mn dyshomeostasis. One example is arginemia, an autosomal recessive disease associated with arginase deficiency caused by a mutation in the ARG1 gene. The result of this disorder is an increase in the concentration of arginine and a periodic increase in the concentration of ammonia in the serum. The symptom of the disease is progressive spastic paraplegia, epilepsy, intellectual disability, and symptoms of encephalopathy [450]. The arginase deficiency may result in liver cholestasis.

Prolidase and prolinase deficiency is also inherited in an autosomal recessive manner. The molecular diagnosis can be established in a proband with suggestive findings and biallelic pathogenic variants in PEPD identified by molecular genetic testing [451]. Prolidase deficiency is characterized by skin lesions (skin ulcers of the lower extremities and telangiectasias of the face and hands), recurrent infections (particularly of the skin and respiratory tract), dysmorphic facial features, variable intellectual disability, and organomegaly (typically splenomegaly but occasionally associated with hepatomegaly) with elevated liver enzymes. Skeletal anomalies, chronic pulmonary disease, anemia, thrombocytopenia, hypergammaglobulinemia, and hypocomplementemia are observed in a minority of affected individuals. An association between prolidase deficiency and autoimmune conditions—particularly systemic lupus erythematosus (SLE)—has been described. This disturbance is accompanied by skin ulcers, mental retardation, splenomegaly, increased urinary proline, and hydroxyproline content, and is associated with impaired manganese metabolism [452]. It has been demonstrated that treatment with manganese and ascorbate, both being prolidase and prolinase cofactors, results in decreased iminopeptiduria and frequency of inflammatory diseases [453].

### 6.2. Mn Accumulation

Familial benign chronic pemphigus (Hailey–Hailey disease) is associated with manganese transporter (SPCA1) gene mutation and manifests with skin blister formation and impaired glycosaminoglycan synthesis [99]. A number of studies have detected a mutation in ATP2C1 gene, which is associated with impaired manganese elimination [454]. ATP132A gene mutation is observed in various parkinsonism types and Kufor–Rakeb syndrome. This gene codes P5-type ATPases [455]. Rare genetic disease related to ACDP1 results in urofacial syndrome (facial grimacing and urinary tract failure). Autosomal recessive disease associated with loss-of-function mutations SLC30A10, a Mn efflux transporter, or SLC39A14, a Mn influx transporter, results in increase Mn levels in blood and brain, and liver. Clinical manifestations of Mn-induced severe neurotoxicity are characterized by liver cirrhosis, muscular dystonia, and polycythemia [195]. Moreover, the mutation of these genes may be observed in patients suffering from Parkinson-like syndromes [393].

According to the KEGG Kyoto Encyclopedia of Genes and Genomes, the pathway of hypermagnesemia with dystonia (HMNDYT) corresponds to ferroptosis (Figure 5). This autosomal recessive disorder of Mn homeostasis is associated with a mutation in two gene families: K14720 solute carrier family 39 (zinc transporter), member 14|(RefSeq) SLC39A14, HCIN, HMNDYT2, and LZ; and K14697 solute carrier family 30 (zinc transporter), member 10|(RefSeq) SLC30A10, HMDPC, HMNDYT1, and Z.

### 6.3. Mn Deficiency and Carcinogenesis

Mn deficiency in humans was described only at the beginning of the 21st century. Clinical signs of Mn deficiency were found in cases of impaired growth and development, bone and cartilage defects, the formation and dysfunction of adipose tissue, impaired glucose tolerance, and infertility. Reduced levels of Mn were detected in epilepsy, osteoporosis, Perthes’ disease, the impaired exocrine function of the pancreas, phenylketonuria, and hemodialysis patients [456].

Studies show that Mn deficiency may be important in carcinogenesis, because it causes the activation of p53 [457], which leads to mitochondria-induced cellular apoptosis, and on the other hand, it breaks down the antioxidant defense system in which both Mn and Mn-dependent SOD are involved [458].

There was a significant relationship between the level of Mn [459] and the polymorphism of the MnSOD gene [460,461], and the risk of breast cancer, especially in perimenopausal women. Obese women, smokers, and those taking hormone replacement therapy are particularly predisposed to breast cancer.

For this malignancy, there are few prognostic biomarkers to monitor the progression of tumor metastasis. MnSOD may be such a predictive biomarker because the expression of its genes changes with the development of the tumor., i.e., it decreases in the early stages of breast cancer and increases in advanced stages. It seems that MnSOD levels act by affecting reactive oxygen species, i.e., superoxide anion radical (O_2_) and hydrogen peroxide (H_2_O_2_), involved in signaling pathways responsible for the proliferation and invasive capacity of angiogenic cancer cells, i.e., breast tumor growth [462]. Regulatory mechanisms of the Mn-axis SOD/AMP-activated kinase (AMPK) is still being studied [462]. In vitro studies, and clinical and epidemiological analyses indicate that the MnSOD/AMPK pathway is responsible for the bioenergetics of cancer cells and is most active in advanced and aggressive breast cancer subtypes [463,464,465].

It was confirmed that the level of gestagens is correlated with the content of Mn and MnSOD. Stimulation with progesterone and the physiological state corresponding to the second phase of the menstrual cycle is characterized by the increased production of MnSOD [466,467].

The effect of hormones on Mn levels is also confirmed by other studies reporting a decrease in Mn levels in postmenopausal women compared to premenopausal women [126]. Similarly, prolactin levels are positively correlated with MnSOD activity [468].

Therefore, in the case of hypophysectomy, which is treated with progesterone, the increased activity of this enzyme is observed. In 2015, a cross-sectional study was conducted on the behavior of blood antioxidant enzymes (SOD, catalase, and glutathione peroxidase), plasma total antioxidant capacity, and oxidative damage (lipid oxidation and protein carbonyl levels) in premenopausal and postmenopausal women without hormone replacement therapy and in women using estrogen (ET) or estrogen–progestogen therapy (EPT) [469]. The study was conducted on a group of 52 women.

The results of the study confirm that the activity of CuZnSOD and MnSOD as well as total plasma antioxidant power were significantly higher in postmenopausal women undergoing EPT than in women in the control group. Moreover, the duration of HRT was positively correlated with the activity of SOD isoforms and total plasma antioxidant power.

The authors of the study unequivocally state that the activity of MnSOD and CuZnSOD in the blood increases while improving the total antioxidant capacity of the plasma in postmenopausal women who use hormone replacement therapy (HRT). However, there are conflicting studies showing that estrogen use does not affect Mn status in adult women [155], and that HRT does not affect serum Mn levels in postmenopausal women [470].

## 7. Metabolic Diseases

Metabolic syndrome (MetS) includes type 2 diabetes (T2DM), obesity, insulin resistance, atherosclerosis, hyperlipidemia, and non-alcoholic fatty liver disease (NAFLD). The main markers of MetS include abdominal obesity, impaired carbohydrate metabolism, elevated blood pressure, and dyslipidemia, which consists of high triglycerides and low HDL (high-density lipoprotein) [471]. Most researchers confirm the relationship between MetS and oxidative stress and inflammation [472]. The relationship between MetS and Mn intake and blood Mn levels has been investigated at the beginning of the 21st century [473]. Epidemiological studies conducted in China showed that a lower dietary intake of Mn was associated with a lower risk of developing MetS [34,36]. Another study confirmed this association, but found that blood and urinary Mn levels were not significantly affected by MetS markers [474]. The association between dietary and environmental Mn exposure, and MetS risk was described recently in a review paper [475].

### 7.1. Type 2 Diabetes Mellitus/Insulin Resistance

T2DM is a state of hyperglycemia that can be the result of insulin resistance or abnormal insulin secretion [476]. High glucose levels cause oxidative stress that can impair islet beta cell function [477], the activation of protein kinase C, and the formation of advanced glycation products [478]. The reduction of ß-oxidation impairs insulin signaling, leading to ectopic lipid accumulation and the induction of insulin resistance [479]. The main biometals acting as cofactors of human metalloproteins directly involved in sugar metabolism and glycosylation processes are Ca, Mg, Mn, Zn, and Co. Studies on rats have confirmed that Mn has an effect on insulin release and gluconeogenesis [480]. Carbohydrate metabolism is compromised in both Mn deficiency and excess. An increase in Mn intake has a positive effect in situations of dietary stress, as it increases insulin secretion, reduces oxidative stress, and prevents endothelial dysfunction [481]. The relationship between blood Mn levels and T2DM in epidemiological studies is less clear. There has been either a positive association [482], a negative association [483] or no association [484]. In 2011, a study was reported on the relationship between Mn intake and the level of glucose, insulin, and insulin resistance among 573 girls aged 8–13 in Spain [485]. A significant positive correlation was found between Mn intake and the homeostasis model assessment (HOMA) levels. The authors of the study suggest that ensuring an adequate supply of Mn in children can prevent both insulin resistance and type 2 diabetes in the future. A few studies show the relationship between Mn levels and glucose metabolism dysfunction in the elderly [484,486]. In 2016, a cross-sectional study was reported that included 2402 Chinese adults over the age of 60 [487]. The relationship between serum Mn levels and the occurrence of pre-diabetes and diabetes was studied. Statistical significance was achieved only for the group of men, where the lowest incidence of diabetes correlated with a moderate range of serum Mn (*p* < 0.05). In addition, the results indicate that there is an inverse linear relationship between Mn levels and pre-diabetes among older women. This is the first study that has attempted to assess the relationship between serum Mn concentration and prediabetes depending on gender. Other studies on the relationship between Mn and diabetes are inconsistent. Some reports describe no statistically significant association between Mn levels and diabetes [488,489,490,491]. In turn, several other studies prove the existence of an inverse [488,492,493,494,495] or positive [482,496,497,498] relationship between the level of Mn and diabetes. Eshak et al. [499] observed strong inverse associations between dietary Mn intake and the risk of type 2 diabetes in women but not men. The participants were 19,862 Japanese men and women. The authors calculated the odds of the 5-year cumulative incidence of type 2 diabetes according to quartiles of dietary manganese intake.

The recently published study from 2022 [500] has explored the relationship between Mn and diabetes. The study was conducted in a population of 2575 hypertensive individuals recruited in China. The authors examined the risk of diabetes. It appeared that the prevalence of diabetes was 27.0% in the studied population. Additionally, a U-shaped association between serum manganese and diabetes was observed in the population with hypertension, and the association was significantly modified by sex. There seems to be a U-shaped relationship between plasma Mn and T2DM, i.e., T2DM is more likely to occur at low and high Mn levels [501,502]. There is a positive correlation between urinary Mn levels and T2DM [60,63] and no correlation between hair, tear, and lymphocyte Mn, levels and T2DM [503,504].

### 7.2. Osteoporosis

IGF-insulin-like growth factor is a polypeptide in the liver that is very similar to insulin. IGF-1 encourages growth in children, and supports the formation of collagen normal bone formation. Mn regulates IGF-1 activity in the tested cell lines and in Mn-deficient rats [505]. A diet low of Mn leads to a decrease in circulating IGF-1 or insulin [506]. The special role of Mn in bone mineralization and the formation of cartilage and collagen has been confirmed in studies [507]. Mg, Zn, and Mn as well as vitamin D3 play a critical role in the development of osteoporosis [508]. Patients with osteoporosis have low blood levels of Mn. The lack of Mn in the diet causes osteopenia due to an imbalance between osteoblast and osteoclast activity and reduces glycosyl transerase activity and proteoglycan synthesis [508]. It has been proven that oxidative stress plays a role in the development of osteoporosis.

It has been proven that oxidative stress plays a role in the development of osteoporosis. Polymorphisms of the SOD2 gene have an overwhelming effect on the occurrence of osteoporosis by virtue of manganese deficiency. Single-mapping studies in Asian Indians with nucleotide polymorphisms (SNP) in the mitochondrial manganese superoxide dismutase (SOD2) gene showed that the G allele of rs5746094 (intronic) and the C allele of rs4880 (exonic) were significantly higher in osteoporotic individuals [509]. The SOD1 gene is located on chromosome 21 and is responsible for the phenotypic oxygen stress in Down syndrome. It is likely that the SOD2 gene is also overexpressed and Mn is missing in Down syndrome, contributing to osteoporosis [510,511,512].

### 7.3. Obesity

Obesity, similar to T2DM and insulin resistance, is associated with oxidative stress and ROS production [513]. The overproduction of ROS impairs the antioxidant capacity of adipose tissue, which has been confirmed in mouse models of obesity [514]. An increase in epididymal adipocyte MnSOD was observed in obese mice [515]. The deletion of MnSOD in mouse adipocytes increases mitochondrial fatty acid oxidation, which prevents obesity [516]. Epidemiological studies conducted in China in a group of men have associated higher Mn intakes above 5.12 mg/d with a reduced risk of abdominal obesity and hypertriacylglycerolemia [473]. Polish researchers observed low serum Mn levels in obese adult men [471]. On the other hand, obese children in the US National Health and Nutrition Examination Survey of over 5000 children had high blood Mn levels [517].

### 7.4. Atherosclerosis

Oxidative stress and inflammation play a key role in the pathogenesis of atherosclerosis [518,519]. Atherosclerosis is characterized by the accumulation of oxidized low-density lipoprotein (oxLDL) and endothelial dysfunction in the arterial wall [520]. There is a confirmed link between MnSOD activity and atherogenesis [521], as MnSOD protects against endothelial dysfunction. Some researchers suggest supplementation with Mn, which has anti-inflammatory effects and attenuates monocyte adhesion to endothelial cells [522]. Although there was no difference in Mn content between atherosclerotic and non-atherosclerotic aortic tissue [523], blood Mn levels were higher in patients with atherosclerosis compared with controls [524].

### 7.5. Non-Alcoholic Fatty Liver Disease

Triglyceride (TG) accumulation in the liver (NAFLD) most commonly leads to inflammatory non-alcoholic steatohepatitis (NASH), followed by fibrosis and cirrhosis [525] NASH, like other metabolic diseases, is associated with mitochondrial dysfunction and oxidative stress [526,527] An in vitro study on human SMMC-7721 cells did not confirm the relationship between Mn levels and NAFLD [528]. Studies in rats suggest that MnSOD mimetics may be effective in the treatment of NASH [529].

## 8. Fertility

### 8.1. Male Fertility

Mn affects reproduction because it is responsible for the activity of genes that regulate hypothalamic gonadotropin-releasing hormone (GnRH) [530,531], and as a cofactor of mevalonate kinase and farnesyl pyrophosphate synthase, it is involved in the synthesis of cholesterol, a precursor of sex hormones [532]. Mn deficiency is associated with a decrease in testosterone levels, which affects male fertility [533,534,535,536].

Research into the effects of Mn on reproduction has mainly been carried out in laboratory animals [537]. Several reports have experimentally confirmed the beneficial effect of Mn supplementation at 150 µM and 200 µM Mn^2+^ on bovine sperm quality during cryopreservation and on the efficacy of artificial insemination [538]. In this study, Mn primarily protected spermatozoa against damage caused by lipid peroxidation (LPO). The antioxidant effect of Mn has also been confirmed by another study [539], in which an improvement in semen parameters in mice, i.e., sperm count and vitality, was observed after the intraperitoneal administration of Mn salt solution.

It should be emphasized that the beneficial effect of Mn on reproduction is observed at its low concentrations because at higher concentrations, it is toxic. The degree of toxicity depends on the dose and duration of exposure to Mn [540]. Epidemiological studies confirm the detrimental effect of high concentrations of Mn exceeding 19.40 µg L^−1^ in serum on the morphology and motility of human spermatozoa [541,542]. Analyzes of the semen of two thousand infertile men showed a significantly higher serum Mn level compared to the group of fertile men [543]. Other epidemiological studies have also indicated impotence and decreased libido in men occupationally exposed to Mn [544,545]. Changes in the levels of sex hormones in the group exposed to Mn compared to the control group, i.e., higher levels of GnRH and luteinizing hormone (LH) and lower levels of testosterone, were confirmed. The experimental data in vitro have shown that Mn interferes with several steps in the testosterone biosynthetic pathway in Leydig cells. Mn acts mainly by reducing the level of the StAR protein, which is responsible for the transfer of cholesterol through the mitochondrial membrane, where the P450scc enzyme initiates the synthesis of all steroid hormones [510].

### 8.2. Female Fertility

Mn is essential for reproductive function as it is a co-factor for the enzymes mevalonate kinase, geranyl pyrophosphate synthetase, and farnesyl pyrophosphate synthase, which are required for cholesterol synthesis. This, in turn, is used for the synthesis of steroid hormones, including progestins, androgens, and estrogens. A diet low in Mn may increase the risk of anovulation, or the inability of the ovary to release an egg during the menstrual cycle, which can lead to infertility. A 2018 study published in the British Journal of Nutrition confirms that low mineral intake is associated with the risk of anovulation in healthy women [545]. The study involved 259 women between the ages of 18 and 44. By measuring levels of the hormones estradiol, progesterone, LH, follicle-stimulating hormone (FSH), sex-hormone-binding globulin, and testosterone, it was found that Mn intake <1.8 mg (RR 2.00, 95% CI 1.02, 3.94) increased the risk of anovulation.

Pregnant women are characterized by a significant increase in serum Cu and Mn levels of 40% (*p* < 0.001) and 16% (*p* = 0.043), respectively, compared to controls [546]. Mn levels during pregnancy were studied in a population of over 1300 women at the Johns Hopkins University Bloomberg School of Public Health [547]. It was found that low levels of Mn in early pregnancy were associated with a higher risk of pre-eclampsia in late pregnancy [548]. The functions of Mn in reproduction are described in the review [549].

In addition, in vitro studies using human placental lobules show that trace elements consumed by pregnant women are actively transported across the placenta to the developing fetus [550]. Mn, along with Zn and Cu, preferentially accumulates in the conceptus, confirming its importance in maintaining pregnancy and fetal development [507]. In studies on pigs, the highest concentrations of Mn were found in tissues rich in mitochondria, bone, and cartilage [551]. The preferential uptake of Mn by the corpus luteum (CL) may be due to its involvement in metabolism or luteal function. The results of studies in farm animals indicate the importance of Mn deficiency in relation to other essential elements. Fertility in cattle is reduced when dietary Mn is below 40 ppm and the Ca:Mn ratio is greater than 4:1. This is reasonable because Mn is a cofactor of enzymes involved in the synthesis of cholesterol, the precursor of progesterone (P4) [552].

In human milk, Mn is present at approximately 10 μg L^−1^ in a form that is bound to lactoferrin [552]. There is therefore no risk of Mn deficiency in breastfed infants.

## 9. Wound Healing

The beneficial effects of Mn and other trace elements on wound healing have been known for over 20 years. Tenaud et al. [553,554] demonstrated in vitro that Mn induces integrin expression in both the proliferative and differentiated stages of wound healing. In the past, thermal waters containing trace amounts of B and Mn have been recommended for washing ulcers. Chebassier et al. [555] studied the effect of thermal water from Saint Gervais on the modulation of keratinocytes in vitro. The study showed that better wound healing was the result of an increase in keratinocyte migration. The incubation of keratocytes in a medium containing Mn salts at concentrations between 0.1 and 1.5 mg/mL accelerated wound healing by 20%.

Skin cancer is mainly treated by surgical resection, chemotherapy, or radiotherapy. Another treatment option is photothermal therapy (PTT) [556]. Cu, Mo, Fe, and Ti compounds have been used as photodynamic agents (PTA). Nanospheres, FeMn(SiO_4_) have been proposed as probes for cancer diagnosis [557] but have also been tested as photothermal agents. The presence of Mn ions increases the thermal vibration of the lattice and improves the photothermal performance. They also protect against oxidative damage and modulate inflammation. A hydrogel containing iron and manganese silicate (FeMn(SiO_4_)) with the addition of sodium alginate (SA) has been described as an effective dressing component for the treatment of wounds following the resection of a skin cancer tumor [558]. Not only was accelerated wound healing achieved, but photothermal therapy of the remaining cancer cells was also possible.

Multi-drug resistant (MDR) bacteria are a problem in wound healing. Dopamine-based Mn-doped hollow carbon nanospheres (MnOx/HNCSs) have been developed as nanozymes and photothermal agents for MDR treatment [559]. Hybrid nanosphere MnOx/HNCSs have enzymatic activity similar to SOD, oxidase, catalase, which leads to the formation of free radicals. They also have a stable photothermal activity. These properties provide antibacterial activity against MDR pathogens. Recently, alginate (Alg) and bacterial nanocellulose (BNC) have become increasingly popular as dressing materials. These biopolymers provide moisture and mechanical strength to composite dressings. Antibacterial properties have been achieved by cross-linking with metal cations such as Mn, Co, Cu, Zn, Ag, cerium (Ce) [560].

Analysis of the chemical composition of acai extracts revealed significant amounts of metals, such as Ca, Mg, Mn, Fe, Zn and Cu, in addition to polyphenols [561]. The extracts were found to have a high migratory activity in human fibroblast cells. The authors suggest the potential of the extracts as wound healing agents.

## 10. Conclusions and Future Prospective

This article reviews the progress of research into the physiological functions of Mn and the effects of Mn dysregulation in humans. Our attention has been focused on studies of Mn as a “structural” essential trace mineral. From this point of view, issues such as the source of exposure to Mn, the dietary intake of Mn, the assessment of Mn status, and the ranges reported in the literature for Mn in different biological matrices, were discussed. In addition to the involvement of Mn in many different physiological processes, the hazards associated with Mn accumulation and the interactions of Mn with other metals are discussed. In order to understand the relationship between exposure to Mn and disorders of brain function and neurodevelopment, fertility, cancer, and metabolic or infectious diseases, it was necessary to collect the current state of knowledge on the role of Mn in oxidative stress, the neurotransmitter mechanism of action, Mn-dependent enzyme activity, protein aggregation, and smooth muscle contraction.

Concluding remarks:

As a component and activator of many enzymes, Mn levels in cells must be tightly controlled by homeostatic mechanisms. The situation is complicated by the fact that other trace metals compete with Mn for the same metabolic pathways, and the range of optimal Mn requirements is very narrow. An imbalance in metal resources at the cellular level disrupts energy conversion processes and causes mitochondrial dysfunction.

Toxicity resulting from excess Mn is a complex process that is not fully understood and includes not only effects on the homeostasis of other metals, such as Fe, Zn, and Ca, but also the dysregulation of glutamate transport or the impairment of dopaminergic function.

Excess Mn is particularly toxic to the brain, which preferentially absorbs manganese. Many mechanisms have been described for the pathogenesis of neurodegenerative diseases in which Mn is involved. However, most of these chronic diseases, such as Alzheimer’s disease, Parkinson’s disease, and Huntington’s disease, still have no effective treatment.

Exposure to Mn in the prenatal and childhood environment is particularly dangerous for the neurological development of children. An association has been observed between Mn and other toxic metals, such as Pb, As, and Cd, and cognitive and behavioral development in children and adolescents.

Future directions:

There is little research on the effects of interactions with organic and inorganic xenobiotics, especially co-exposure to multiple metals with an analysis of the antagonism or synergy of their action, which can lead to increased neurotoxicity. It is advisable to address the aspect of competition of Mn with other metals, including Fe, Mg, Zn, Cu, Pb, and Ca.

Another direction with potential for development is the use of Mn in hybrid materials to aid the treatment of antibiotic-resistant wounds and infections, as well as anti-cancer photothermotherapy. Increasing attention is also being paid to the role of Mn in the development of metabolic diseases and obesity.

The widespread presence of Mn in natural, anthropogenic, and dietary environments, i.e., in airborne particles, drinking water, soil, and vegetables, is a cause for concern, and strategies to prevent overexposure need to be addressed in view of the increasing number of neurodevelopmental and metabolic disorders. Protective strategies against overexposure as well as the continuous monitoring of food and water as the main sources of Mn seem to be crucial.

Relatively little attention has been paid to therapies that can reverse Mn neurotoxicity. An example of such a study is the report by Zhang et al. from 2023, which showed that glutamine supplementation is able to reverse Mn neurotoxicity [562], or the Ling et al. study [414] from 2018, which proposed tyrphostin AG-825 for this purpose.

There is also a need for systematic epidemiological studies and their meta-analyses on the health effects of Mn in children and adults separately.

These issues are beyond the scope of this review and should be the subject of systematic cross-sectional analyses in the future.

## Figures and Tables

**Figure 1 ijms-24-14959-f001:**
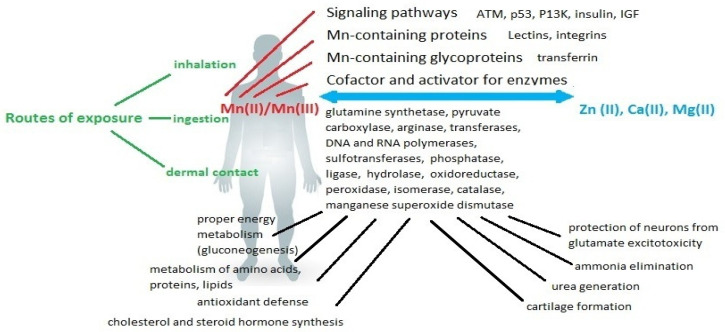
Main exposure routes and physiological functions of Mn in humans.

**Figure 2 ijms-24-14959-f002:**
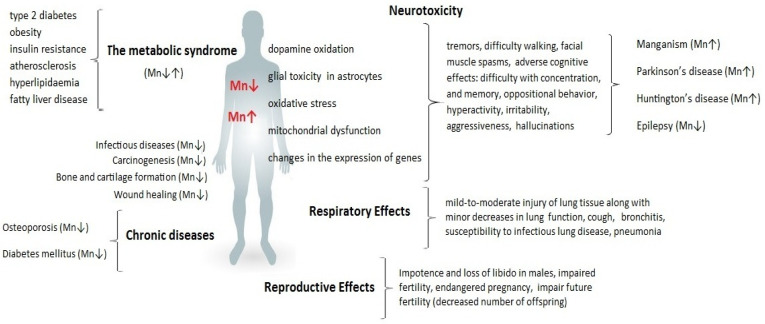
Effects of Mn dysregulation.

**Figure 3 ijms-24-14959-f003:**
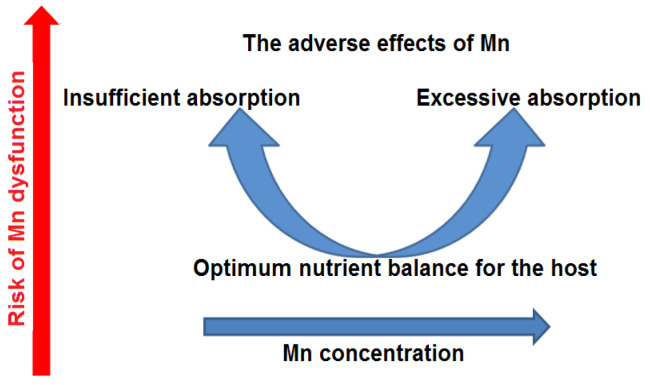
The U-shaped curve for risk associated with Mn status in the body.

**Figure 4 ijms-24-14959-f004:**
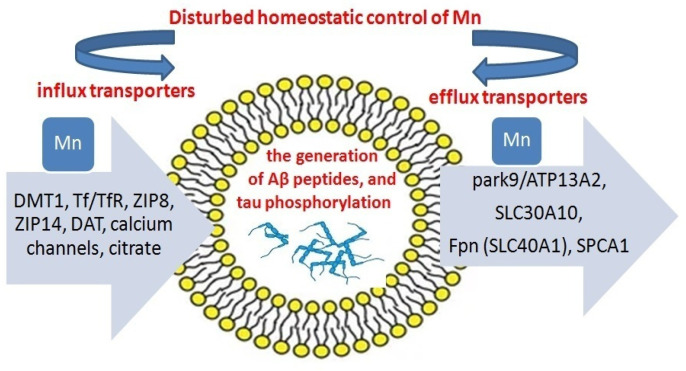
A model of the imbalance in the Mn transport system.

**Figure 5 ijms-24-14959-f005:**
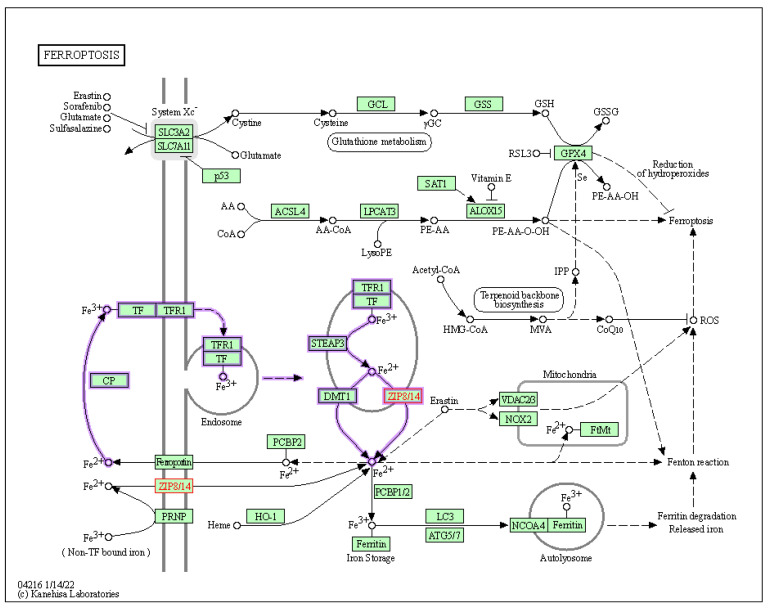
Hypermagnesemia−dystonia KEGG pathway (map:hsa04216) that corresponds to ferroptosis. Fe-Transferrin transport is shown in purple. The transport proteins ZIP8 and ZIP14, which support the transcellular transport of divalent Mn ions within the microvascular capillary endothelial cells (BMVECs) and the blood–brain barrier (BBB), are marked in red (network H01938 Hypermanganesemia with dystonia). (generated online from https://www.genome.jp/entry/H01938) (accessed on 6 March 2023).

**Table 1 ijms-24-14959-t001:** The levels of Mn in different matrices reported in the literature.

Matrix	Mn Concentration	Population	Ages	Location	Method	Ref.
plasma	mean: 14.3 ± 11.4 nmol L^−1^ range: 3–27.5 nmol L^−1^	68	22–75 y	Germany	ETAAS	[91]
	mean: 25.5 ± 22.8 nmol L^−1^ range: 2.7–48.3 nmol L^−1^	129	1 mth–18 y			
serum	range: 4.7–215 nmol L^−1^	584	16–18 y	India	GFAAS	[121]
serum	2.36 µg L^−1^	50	35–70 y	Italy	SEC-ICP-DRC-MS	[40]
blood	mean: 10.240 ± 2.834 range: 6.597–15.740 μg L^−1^	200 (97 f, 103 m)	3–14 y	Kinshasa Congo	ICP-MS	[122]
urine	mean: 0.510 ± 1.643 range: 0.07–1.703 μg L^−1^; 0.948 ± 5.254 μg g^−1^ of creatinine					
blood	female: 1.403 µg/dL male: 1.192 µg/dL	2005	>20 y	Korea	GFAAS	[123]
blood, urine	blood: 10.5–17.9 μg L^−1^ urine: 0.133–0.410 μg L^−1^		3–11 y	USA		[124]
blood, scalp hair, fingernails	blood: 9.95 µg L^−1^ scalp hair: 380.98 ng g^−1^ fingernails: 936.71 ng g^−1^	130 (95 f, 35 m)	20–71 y	Spain	ICP-MS	[125]
blood	pre- menopausal: 14.41 µg L^−1^ post-: 12.92 µg L^−1^	1826 f	20–60 y	Korea	GFAAS	[126]
blood	10–20 wk: 150.4 ± 53.4 nmol L^−1^ 25 wk: 171.6 ± 59.7 nmol L^−1^ 34 wk: 230.0 ± 67.8 nmol L^−1^ neonate: 737.7 ± 209.7 nmol L^−1^	34 pregnant f (wk gestation), 34 neonate	pregnancy period, 3–4 day old	Australia	GFAAS	[127]
blood	pregnant: 2.4 µg/dL nonpregnant: 0.8–1.2 µg/dL	470	14–36 y	Canada	DRC–ICP-MS	[128]
serum	29.32 μg L^−1^	1183 (890 m, 293 f)	18–65 y	Brazil	AAS	[129]
hair	mean: 0.320 mg g^−1^ range: 0.239–0.789 mg g^−1^	260 (122 f, 138 m)	30–49 y	Kazakhstan	ICP-MS	[130]
blood, serum	blood: 9.03 ± 2.25µg L^−1^ serum: 1.82 ± 0.64µg L^−1^	80 (34 f, 26 m)	18–64 y	USA	ZAAS, NAA	[131]
hair	males: 0.59 ± 0.61 µg g^−1^ females: 1.09 ± 1.43 µg g^−1^ mean: 0.95 ± 1.27µg g^−1^ range: 0.29–1.76 µg g^−1^	7256 (5161 f, 2095 m)	20–60 y	Russia	ICP-MS	[132]
blood	range: 1.5–22 µg L^−1^ mean: 7.40µg L^−1^	206	16–70 y	United Kingdom	ETAAS, ICP-MS	[47]
urine	range: 0.09–7.8µg L^−1^ mean: 0.3 µg/L	188				
blood	range: 7.0–14.1 µg L^−1^ median: 9.8 µg L^−1^	99	<5 y (*n* = 17) 5–15 (*n* = 31) 15–18 (*n* = 20)	France	ICP-MS	[133]
plasma	range: 0.53–2.21µg L^−1^ median: 0.97µg L^−1^	96				
blood	range: 5.9–13.3 µg L^−1^	106	25–55 y	France	ICP-MS	[134]
plasma	range: 0.35–1.08µg L^−1^					
plasma	range: 3.0–27.5 nmol L^−1^ mean: 14.3 nmol L^−1^	68	22–75 y	Germany	ETAAS	[93]
	range: 3.0–53 nmol L^−1^ mean: 25.5 nmol/L	129	1 mth-18 y			
blood	range: 4.8–18 µg L^−1^; mean: 9.0 µg L^−1^	130 (80 f, 50 m)	18–70 y	Germany	ICP–MS	[135]
blood, serum	blood: 144 ± 43.6 nmol L^−1^ serum: 28.4 ± 21.0 nm L^−1^	1016	70 y	Sweden	ICP-SFMS	[136]
hair	0.33 ± 0.16 µg g^−1^	2	-	Brazil, Thailand	LA-ICP-MS, ICP-MS	[137]
urine	0.46 µg L^−1^	132 (82 m, 50 f)	18–66 y	UK	ICP-MS	[53]
blood	range: 6.9–18.4 µg L^−1^ mean: 9.6 µg L^−1^	1125 (506 m, 619 f)	18–60 y	Brazil	ICP-MS, ETAAS	[138]
blood	median: 10.4 μg L^−1^	100 (64 f, 36 m)	36 mths	Congo	ICP-MS	[139]
urine	5.2 ± 0.7µg L^−1^	35	6–11 y	Mexico	ICP-OES	[140]
plasma	range: 0.63–2.26 µg L^−1^	40 (9 m, 11 f)	5–18 y	Germany	ICP-MS, ICP-AES	[141]
urine	range: 0.427–0.761 µg L^−1^	1022 (460 m, 541 f)	18–80 y	Belgium	ICP-MS	[45]
blood	range: 5–12.8 µg L^−1^ median: 7.6 µg L^−1^	100	-	Canada	ICP-MS	[117]
urine	range: 0.11–1.32 µg L^−1^ median: 0.31 µg L^−1^	100				
plasma	range: 0.63–2.26 µg L^−1^ median: 1.12 µg L^−1^	100				
hair	range: 0.016–0.57 ng mg^−1^ median: 0.067 ng mg^−1^	45				

Abbreviations: ETAAS, electrothermal atomic absorption spectrophotometry; GFAAS, graphite furnace atomic absorption spectrophotometry; SEC-ICP-DRC-MS, size-exclusion chromatography coupled to inductively coupled plasma mass spectrometry dynamic reaction cell; ZAAS, Zeeman effect flameless atomic absorption spectrophotometry; NAA, neutron activation analysis; ICP-SFMS, inductively coupled plasma-sector field mass spectrometry; LA-ICP-MS, laser ablation inductively coupled plasma mass spectrometry; ICP-AES, inductively coupled plasma atomic emission spectroscopy; y, years; wk, week; mth, month; m, male; f, female.

**Table 2 ijms-24-14959-t002:** Studies assessing the connection between exposure to Mn and neurodevelopmental disturbances in children.

Neurodevelopmental Disturbances	Biomarkers	Population	Major Conclusion	Ref.
IQ Decrement	hair, blood	infants and toddlers	higher hair Mn concentrations are associated with lower IQ scores; blood biomarkers give inconsistent findings	[299]
Cognitive Functions	blood, hair, air	perinatal and childhood exposure (38 children from USA (near the Ohio River)	Mn exposure has negative impacts on cognition and behavior	[300]
	umbilical cord serum	933 mother–newborn pairs in Shanghai, China	above 5 μg L^−1^ in three-day-olds showed cognitive deficits in NBNA test	[301]
	blood	6 to 12 months, 448 children born in Mexico City	U-shaped association between Mn level and mental development scores on the BSID	[302]
	cord blood	2-year-old children in Taiwan (230 pairs)	Mn level was associated with decrements in cognitive and language subscales of CDIIT	[303]
Intellectual Ability	blood and hair	children aged 7–9 years, (*n* = 404) from Marietta and Cambridge, Ohio	both low and high Mn concentrations in blood and hair were negatively associated with the total IQ scores	[23]
	blood	children 8–11 years, 1089 children living in in South Korea	high Mn is associated with lower scores in thinking, reading, calculations, and LQ in the LDES and a higher commission error in the CPT; low Mn is associated with lower color scores in the Stroop test	[304]
	hair	children aged 14–45 months (*n* = 60) from Pennsylvania, USA	no evidence of an association between Mn concentrations and BSID scores in a region of low-level Mn exposure	[305]
	blood, drinking water	prenatal and early childhood (2–3 year old), 524 children from the Sirajdikhan and Pabna Districts of Bangladesh	Mn content in water was associated with fine motor scores in an inverse U-relationship, the adverse effects of Mn was observed in the case of lower Pb level	[306]
	umbilical cord blood samples	in utero exposure, 2 years of age (230) from Taiwan	Mn above the 75th percentile had a significant adverse association with the overall, cognitive, and language quotients of the CDIIT	[303]
	blood	adolescents aged 11–14 years (*n* = 299) from USA	Mn was not associated with cognitive and behavioral outcomes at the low exposure levels	[307]
	drinking water	from fetal life to school age (5, 10 years), 1265 children in rural Bangladesh	none of the Mn exposures was associated with the children’s cognitive abilities; prenatal Mn exposure was positively associated with cognitive function in girls, boys were unaffected	[308]
Executive Function	blood, hair, drinking water	children aged 6–12 years old (*n* = 63) from Brasil	blood Mn was associated with visual attention, negatively associated with visual perception and phonological awareness; hair Mn was inversely associated with working memory; Mn from drinking water associated with inhibition of written language and executive functions	[309]
	hair	7- and 12-year-old children (*n* = 70) living near a ferromanganese alloy plant	airborne Mn exposure may be associated with lower IQ and neuropsychological performance in tests of executive function of inhibition responses, strategic visual formation, and verbal working memory in TAVIS-3R, WCST, WISC-III Digit Span subtest and Corsi Block	[109]
Memory	hair and blood	children aged 7–11 years, 174 children living in the Molango State of Mexico	Mn levels showed a negative association with the CAVLT, WRAML scores	[108]
Academic Achievement	drinking water	children aged 8–11 years, 840 children living in Araihazar, Bangladesh	no significant relation was observed	[310]
Effects on Motor Functions	drinking water	children aged 20–40 months, 524 children, in the Sirajdikhan and Pabna Districts of Bangladesh	Mn < 400 μg/L, Mn is beneficial to fine motor development, whereas at Mn > 400 μg/L, Mn exposure is detrimental for motor function	[306]
	blood	children ages 7–9 years, 55 children residing in Marietta, Ohio, USA	Mn exposure was significantly associated with poor postural balance	[298]
	hair, blood	7 and 11 years old, 195 children (100 exposed and 95 unexposed) from Mexico	negative association of Mn exposure on motor speed and coordination was shown	[311]
Behavior	blood	children/adolescent	association between early life exposure to Mn and children/adolescent behavior	[312]
	blood	5–15 years, 92 children from rom Al Ain Educational Zone, UAE	Mn was significantly associated with ADHD	[313]
	prenatal and postnatal enamel regions of deciduous teeth	84 (aged 9–14 years) Caucasian children	no significant differences in Mn level for children with ASDs compared with TD children, children with ASDs have slightly lower Mn levels, no significant differences between children with HDB and TD children	[314]
	drinking water	8–11 years (201 children) in Bangladesh	positive dose–response relationship between Mn level and CBCL-TRF total scores	[315]

Abbreviations: Intelligence quotient (IQ); the Learning Disability Evaluation Scale (LDES); the Child Behavior Checklist (CBCL); continuous performance test (CPT); the learning quotient (LQ); the Bayley Scales of Infant Development III (BSID); the Comprehensive Developmental Inventory for Infants and Toddlers (CDIIT); The Wechsler Intelligence Scale for Children (WISC); the Conners–Wells’ Adolescent Self-Report Scale Long Form (CASS:L); the Strengths and Difficulties Questionnaire (SDQ); Test of Visual Attention (TAVIS-3R); cognitive flexibility (WCST); verbal and visual working memory (WISC-III Digit Span subtest and Corsi Block); the Neonatal Behavioral Neurological Assessment (NBNA); Bayley Scales of Infant Development (BSID); high levels of disruptive behavior (HDB), typically developing (TD) children; the standardized Child Behavior Checklist-Teacher’s Report Form (CBCL-TRF).

## Data Availability

Not applicable.

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
