# Peer review of "Consequences of Disturbing Manganese Homeostasis"

_ijms, 2023, doi:10.3390/ijms241914959_

Round 1

Reviewer 1 Report

The manuscript represents a comprehensive review of the physiological and toxicological functions of manganese. I have recommendations for minor revisions.

1. Lines 44 and 45 describe macroelements that are essential to life and biological processes but in lines 46 - 50, a mixture of trace elements that are essential or toxic. The following paragraphs discuss the physiological roles of these elements, so this mix of elements may confuse the reader. Bromine (Br), silicon (Si), cesium (Cs), aluminum (Al), lead (Pb), cadmium (Cd), Rubidium (Rb) have either no known nutritionally relevant role and contribute either neutral or toxic physiological effects. These elements should be deleted from the statement in lines 46-50.

2. Lines 70-72 contain incorrect statements that need to be corrected. Se is not found in soil as either a silicate or a sulfide, but rather in the forms of selenides and selenates. Fe is not found in the native elemental form but rather in ferrous or ferric forms as sulfides or oxides.

3. The statements in lines 77-79 show lack of understanding of inorganic biochemistry. Some transition metals are predominantly found as complexes in the body, but not predominantly as complexes of the compounds listed. Most alkali metals (the highest concentration metal elements) do not occur as complexes at all, but rather as hydrated ions. The authors should delete the statements in lines 77-79.

4. In lines 137-138, the oxidation states of manganese include some that do not occur in nature, but only in oxygen-free, moisture free vessels of synthetic inorganic chemists. In particular, none of the negative, 0, or +1 oxidation states of manganese exist in the body. The +6 and +7 oxidation states are strongly oxidizing and toxic as a result. The authors should limit the statement to pointing out that the II and III oxidation states of manganese are the most physiologically relevant states. Perhaps they might also mention the VI and VII oxidation states only if they also state that they are strongly oxidizing and toxic.

4. Lines 168-170 should be deleted. There are no volatile naturally occurring forms of manganese that would occur in water vapor, especially during bathing, and definitely not enough to cause brain damage.

5. Line 174: "The source of exposure to Mn is drinking water" should be corrected to state, "A major source of exposure to Mn is drinking water."

6. Line 187 states a lack of reference ranges for manganese biomonitoring, but the authors list one of the best biomonitoring sources in citation 121, the NHANES National exposure report. Blood, urine, and creatinine corrected urine manganese reference ranges can be found specifically at https://www.cdc.gov/exposurereport/data_tables.html .

7. Lines 189-190: The authors incorrectly state that mercury, nickel, and chromium are only monitored in cases of exposure, but they are all monitored in the NHANES exposure reports.

8. Line 319: Deletion of the words, "level rather than" will make the statement correct.

9. Line 440: Replacement of "the cause of cell death" with "a cause of cell death" will make the statement correct.

10. Line 564: TNF- should be TNF-α.

11. Line 565: NF-B should be NF-κB.

12. Lines 788-789: Glutathione-S-transferase is an enzyme that catalyzes nucleophilic attack of the sulfur from glutathione on hydrophobic organic molecules like chlorodinitrobenzene so that it will become soluble attached to glutathione and more easily excreted. Glutathione sulfur binds soft acid cation metals as a ligand without the use of the Glutathione-S-transferase enzyme. Please eliminate the discussion of glutathione-S-transferase related to metals and leave only the mention of glutathione binding certain metals such as mercury, cadmium, etc.

Author Response

The manuscript represents a comprehensive review of the physiological and toxicological functions of manganese. I have recommendations for minor revisions.

All the authors of the paper would like to thank the reviewer for his insightful review. We hope that, thanks to his comments, we have been able to correct our errors and omissions. Thanks again.

  1. Lines 44 and 45 describe macroelements that are essential to life and biological processes but in lines 46 - 50, a mixture of trace elements that are essential or toxic. The following paragraphs discuss the physiological roles of these elements, so this mix of elements may confuse the reader. Bromine (Br), silicon (Si), cesium (Cs), aluminum (Al), lead (Pb), cadmium (Cd), Rubidium (Rb) have either no known nutritionally relevant role and contribute either neutral or toxic physiological effects. These elements should be deleted from the statement in lines 46-50.

Answer: Of course, we agree with the reviewer that the above elements should be removed from this list. The sentence has been corrected accordingly.

  1. Lines 70-72 contain incorrect statements that need to be corrected. Se is not found in soil as either a silicate or a sulfide, but rather in the forms of selenides and selenates. Fe is not found in the native elemental form but rather in ferrous or ferric forms as sulfides or oxides.

Answer: We fully agree that this information should be detailed. The corrections have been made and marked in red.

  1. The statements in lines 77-79 show lack of understanding of inorganic biochemistry. Some transition metals are predominantly found as complexes in the body, but not predominantly as complexes of the compounds listed. Most alkali metals (the highest concentration metal elements) do not occur as complexes at all, but rather as hydrated ions. The authors should delete the statements in lines 77-79.

Answer: Yes. We agree that this part also needs more explanation. We have added the following in place of this imprecise statement

“In the body, alkali metals such as Na+, K+ occur as hydrated ions with high mobility. These ions have weak ligand binding strengths that are predominantly ionic in origin. Mg2+ and Ca2+ have intermediate binding strengths to organic ligands, most commonly coordinated by oxygen donors derived from glutamic acid, aspartic acids, carbonyl oxygens, peptides and water. In turn, transition metal elements containing electrons in the d-shell form metal complexes with strongly bound ligands. For example, Fe, which is mostly bound in haemoglobin, ferritin, myoglobin and transferrin, is also a component of many enzymes, e.g. catalase, oxidases, peroxidases, dehydrogenases, cytochrome or nucleotide reductase; Cu, which is bound to several proteins (cerebrocuprein, erythrocuprein and hepatocuprein) and enzymes (tyrosinase, cytochrome oxidase, ascorbic acid oxidase, uricase, ceruloplasmin, superoxide dismutase, amine oxidase, dopamine hydroxylase); Zn, which is an essential component of many metalloenzymes (carbonic anhydrase, alkaline phosphatase, pancreatic carboxy-peptidases, erytosoly superoxide dismutase, relinene reductase).”

  1. In lines 137-138, the oxidation states of manganese include some that do not occur in nature, but only in oxygen-free, moisture free vessels of synthetic inorganic chemists. In particular, none of the negative, 0, or +1 oxidation states of manganese exist in the body. The +6 and +7 oxidation states are strongly oxidizing and toxic as a result. The authors should limit the statement to pointing out that the II and III oxidation states of manganese are the most physiologically relevant states. Perhaps they might also mention the VI and VII oxidation states only if they also state that they are strongly oxidizing and toxic.

Answer: Yes. We agree with the review. Thank you for this suggestion. We corrected this part into:

“Mn has a high redox activity. There are eleven known oxidation states of Mn ranging from -3 to +7 with different coordination geometries, of which Mn2+, Mn3+ are the most physiologically relevant states. Mn2+ is the stable and more easily degraded form in bio-logical systems. However, it is important to note that Mn3+, complexed with transferrin, is a more potent oxidant that is even more stable than Mn2+. It should be noted that Mn in the oxidation state Mn6+ and Mn7+ are highly oxidising and therefore toxic.”

  1. Lines 168-170 should be deleted. There are no volatile naturally occurring forms of manganese that would occur in water vapor, especially during bathing, and definitely not enough to cause brain damage.

Answer: We found the following works: Elsner RJ, Spangler JG. Neurotoxicity of inhaled manganese: public health danger in the shower? Med Hypotheses. 2005;65(3):607-16. doi: 10.1016/j.mehy.2005.01.043. PMID: 15913899; Aschner M. Manganese in the shower: mere speculation over an invalidated public health danger. Med Hypotheses. 2006;66(1):200-1. doi: 10.1016/j.mehy.2005.08.002.  According to above discussion we mentioned them in our work. Definitely our statement appears to be too strong. We have improved our description in this way:

“According to research by scientists at Wake Forest University in the USA, even long-term exposure to Mn-containing water in the shower may pose a risk of central nervous system (CNS) neurotoxicity [30]. The authors of the study based their findings on the assumption that intranasally administered Mn bypasses the blood-brain barrier and enters the CNS directly via the olfactory route, as demonstrated in animal studies. According to the authors, even 10 years of bathing in Mn-contaminated water results in exposure to aerosol Mn at doses many times higher than those that cause Mn accumulation in the brain of rats (3 and 112 times higher). Aschner [31] pointed out that the above study lacked information on the solubility of Mn in water, the size of the particles in the air aerosol and its clearance. The authors also did not determine whether it is possible to reach the threshold concentration of Mn toxicity, which is above 100 µM in cells, by taking a shower every day.”

  1. Line 174: "The source of exposure to Mn is drinking water" should be corrected to state, "A major source of exposure to Mn is drinking water."

Answer: It was corrected accordingly.

  1. Line 187 states a lack of reference ranges for manganese biomonitoring, but the authors list one of the best biomonitoring sources in citation 121, the NHANES National exposure report. Blood, urine, and creatinine corrected urine manganese reference ranges can be found specifically at https://www.cdc.gov/exposurereport/data_tables.html .

Answer: Thank you for your suggestion. For clarification, we have added the following at this point:

“Geometric means and selected percentiles of measured Mn blood concentrations (in µg/L), urine concentrations (in µg/L), and creatinine-corrected urine concentrations (in µg/g of creatinine) for the U.S. population from the National Health and Nutrition Examination Survey, collected between 2011-2018, are available specifically at https://www.cdc.gov/exposurereport/data_tables.html.”

  1. Lines 189-190: The authors incorrectly state that mercury, nickel, and chromium are only monitored in cases of exposure, but they are all monitored in the NHANES exposure reports.

Answer: Thank you for this suggestion. We corrected this sentence to be more precise in this way:

“In many countries, levels of toxic metals, i.e. mercury (Hg), nickel (Ni) and chromium, are monitored only in the case of occupational exposure. However, they are all monitored in the NHANES exposure reports for the US population.”

  1. Line 319: Deletion of the words, "level rather than" will make the statement correct.

Answer: Thank you. It was corrected.

  1. Line 440: Replacement of "the cause of cell death" with "a cause of cell death" will make the statement correct.

Answer: Thank you. It was corrected.

  1. Line 564: TNF- should be TNF-α.

Answer: Thank you. It was corrected.

  1. Line 565: NF-B should be NF-κB.

Answer: Thank you. It was corrected.

  1. Lines 788-789: Glutathione-S-transferase is an enzyme that catalyzes nucleophilic attack of the sulfur from glutathione on hydrophobic organic molecules like chlorodinitrobenzene so that it will become soluble attached to glutathione and more easily excreted. Glutathione sulfur binds soft acid cation metals as a ligand without the use of the Glutathione-S-transferase enzyme. Please eliminate the discussion of glutathione-S-transferase related to metals and leave only the mention of glutathione binding certain metals such as mercury, cadmium, etc.

Answer: Yes, thank you very much. We have shortened this discussion. Thank you again for a very helpful explanation.

Reviewer 2 Report

The article, titled 'Implications of Disrupting Manganese Balance,' offers a comprehensive summary of the consequences associated with the disturbance of manganese homeostasis, substantiated with appropriate citations of prior research. I would like to provide a few suggestions for improvement:

  1.  Consider incorporating the KEGG pathways to establish connections between manganese imbalance and neurodegenerative diseases.

  2. While the article is well-written, it would greatly benefit from enhancements in the final section, '10. Conclusion and Future Directions.' so that it can reflect the whole article. It is advisable to address the competitive aspects of manganese (Mn) with other metals, including iron (Fe), magnesium (Mg), zinc (Zn), copper (Cu), lead (Pb), and calcium (Ca).

Author Response

The article, titled 'Implications of Disrupting Manganese Balance,' offers a comprehensive summary of the consequences associated with the disturbance of manganese homeostasis, substantiated with appropriate citations of prior research. I would like to provide a few suggestions for improvement:

Answer: All the authors of the paper would like to thank the reviewer for his insightful and inspiring review. We hope that, thanks to his comments, we have been able to improve our work and gain inspiration for the future. Many thanks again.

  1. Consider incorporating the KEGG pathways to establish connections between manganese imbalance and neurodegenerative diseases.

Answer: Thank you for this suggestion. A very interesting proposition that we had not considered before. We will definitely use it in our future works. KEGG's analyzes in this respect have already been described and therefore we have cited these works in the appropriate places with appropriate comments:

Ling J, Yang S, Huang Y, Wei D, Cheng W. Identifying key genes, pathways and screening therapeutic agents for manganese-induced Alzheimer disease using bioinformatics analysis. Medicine (Baltimore). 2018 Jun;97(22):e10775. doi: 10.1097/MD.0000000000010775. PMID: 29851783; PMCID: PMC6392515.

Tinkov AA, Paoliello MMB, Mazilina AN, Skalny AV, Martins AC, Voskresenskaya ON, Aaseth J, Santamaria A, Notova SV, Tsatsakis A, Lee E, Bowman AB, Aschner M. Molecular Targets of Manganese-Induced Neurotoxicity: A Five-Year Update. Int J Mol Sci. 2021 Apr 28;22(9):4646. doi: 10.3390/ijms22094646. 

  1. While the article is well-written, it would greatly benefit from enhancements in the final section, '10. Conclusion and Future Directions.' so that it can reflect the whole article. It is advisable to address the competitive aspects of manganese (Mn) with other metals, including iron (Fe), magnesium (Mg), zinc (Zn), copper (Cu), lead (Pb), and calcium (Ca).

Answer: Yes. We added this aspect to future directions as well as dire need for supplementation that could reverse manganese neurotoxicity [Zhang S, Zhang J, Wu L, Chen L, Niu P, Li J. Glutamine supplementation reverses manganese neurotoxicity by eliciting the mitochondrial unfolded protein response. iScience. 2023 Jun 15;26(7):107136. doi: 10.1016/j.isci.2023.107136].
